# A breakage–replication/fusion process explains complex rearrangements and segmental DNA amplification

Cheng-Zhong Zhang ●[1,2,3] ✉, Carlos Mendez-Dorantes ●[2,3,4],
Kathleen H. Burns ●[2,3,4] & David Pellman ●[3,5,6,7]

Segmental copy-number gains are major contributors to human genetic variation and disease, but how these alterations arise remains incompletely understood. Here, based on the analyses of both experimental evolution and human disease genomes, we describe a general mechanism of segmental copy-number gain from a rearrangement process termed 'breakage–replication/fusion'. The hallmark genomic feature of breakage–replication/fusion is adjacent parallel breakpoints: two or more rearrangement breakpoints derived from replication of a single ancestral DNA end. We show that adjacent parallel breakpoints are a widespread feature of DNA duplications in human disease genomes and experimental models of chromothripsis. In addition to adjacent parallel breakpoints, breakage–replication/fusion also explains two other patterns of complex rearrangements with unclear provenance: chains of short (≤1 kb) insertions and high-level amplification consisting of inverted segments. Together, these findings revise the mechanistic model for chromothripsis and provide a new conceptual framework for understanding the origin of segmental DNA duplication during genome evolution.

Since the pioneering studies by Alfred Sturtevant[1], the impact of gene duplications on cellular[2], organismal[3,4] and disease phenotypes[5,6] has been well-recognized. However, the mutational processes generating duplicated gene copies are incompletely understood. Duplications of DNA sequences are thought to arise from two classes of DNA sequence rearrangements[7,8]. In the first class, duplicated DNA arises from an asymmetric distribution of replicated DNA between daughter cells. One such example is the breakage–fusion–bridge (BFB) cycle[9,10]. In the second class, duplicated DNA arises from extra DNA synthesis (that is, over-replication within a single cell cycle). One proposed mechanism of this class is microhomology-mediated break-induced replication (MMBIR)[11], a type of break-induced replication[12,13] initiated by microhomology-mediated strand invasion (Supplementary Video 1).

Many rearrangement patterns associated with copy-number gains in human disease genomes[14–18] do not fall neatly into either class. A particular puzzling case is chromothripsis[19]. We previously demonstrated that chromothripsis can arise from a reciprocal distribution of shattered chromosome fragments between daughter cells[20,21]. However, this model cannot explain chromothripsis with complex insertions[19–23] and/or multi-copy DNA gains[23–25]. These features were commonly attributed to DNA re-replication[14] or to additional processes of chromosome fragmentation[26–28], but none of these mechanisms has definitive experimental support.

Here, we describe breakage–replication/fusion, a new rearrangement process that directly generates segmental DNA gains after one round of chromosome fragmentation. The breakage–replication/

[1]Department of Data Sciences, Dana-Farber Cancer Institute, Boston, MA, USA. [2]Department of Pathology, Harvard Medical School, Boston, MA, USA. [3]Cancer Program, Broad Institute of MIT and Harvard, Cambridge, MA, USA. [4]Department of Pathology, Dana-Farber Cancer Institute, Boston, MA, USA. [5]Department of Pediatric Oncology, Dana-Farber Cancer Institute, Boston, MA, USA. [6]Department of Cell Biology, Blavatnik Institute, Harvard Medical School, Boston, MA, USA. [7]Howard Hughes Medical Institute, Chevy Chase, MD, USA. ✉e-mail: cheng-zhong_zhang@dfci.harvard.edu

fusion mechanism follows from the recognition that a replisome passing through a free double-strand DNA end produces two different DNA ends, and fusions involving these replicated 'sister' DNA ends can produce DNA duplication. We first demonstrate that replication/fusion of DNA ends generates 'adjacent parallel breakpoints', a widespread feature in cancer genomes. We then show that chromosome breakage–replication/fusion explains three patterns of copy-number gains in human disease genomes that were of unclear provenance: low-level gains of segments >1 Mb, high-level amplifications consisting of inverted duplications, and chains of short insertions (0.1–1 kb) at rearrangement junctions. Together, these findings provide a new conceptual framework for analyzing somatic genome evolution in human disease and other biological contexts.

## Results

We first show how breakage–replication/fusion converts free DNA ends into breakpoints on rearranged sequences and then show how breakage–replication/fusion of chromosome fragments produces segmental copy-number gains and amplifications. We place particular emphasis on distinguishing the genomic feature of a rearranged DNA sequence (for example, breakpoints) from the molecular feature of the ancestral chromosome (for example, DNA ends). See Supplementary Note, Section 1 for the complete list of definitions.

### Rearrangements from breakage–replication/fusion of DNA ends

A DNA double-strand break (DSB) generates two reciprocal DNA ends (Fig. 1a). In the G1 phase, these ends can undergo classical non-homologous end-joining (c-NHEJ): they can be ligated together, creating a rearranged sequence with small deletions (or less frequently, duplications), or ligated to DNA ends from distal sites, creating translocations[29,30]. In either scenario, the ancestral DNA ends are converted to two breakpoints (open circles in Fig. 1a) separated by a small gap, which we term adjacent gapped breakpoints. As the ligation(s) occur before replication, the rearranged DNA sequences are preserved in both sister chromatids after replication. This cascade of events defines the breakage–fusion–replication sequence (Supplementary Video 2).

If the DSB ends have substantial overhangs that prevent c-NHEJ[31,32] (for example, because of 5′-resection[33–35] or 3′-exonuclease degradation[24]), they can remain unligated during G1 and persist into S phase. During S phase, these ends, like broken chromosome ends[36], are replicated to generate two 'sister' DNA ends. Ligations of these replicated DNA ends can generate up to four rearrangement junctions (Fig. 1b). This cascade of events defines the breakage–replication–fusion sequence (Supplementary Video 3). In breakage–replication–fusion, a staggered DNA end is converted into two adjacent but non-identical breakpoints with the same orientation, which we term adjacent parallel breakpoints. When the sister DNA ends are directly ligated to each other, it produces a 'foldback' junction, joining two adjacent parallel breakpoints. Foldback junctions are often assumed to indicate fusions between the ends of broken sister chromatids in BFB cycles[37,38]; later, we will show that such fusions also occur between sister DNA fragments.

In a variation of breakage–replication–fusion, two single-strand DNA (ssDNA) ends with a small gap are converted into two reciprocal DSB ends by replication[39,40] (Fig. 1c). These two DSB ends can generate two rearrangement breakpoints with either a small gap (i) or a small overlap (ii) by a replication bypass mechanism[18,41,42]. We refer to the latter as adjacent overlapping breakpoints.

A single DSB end undergoes either breakage–fusion–replication or breakage–replication–fusion. However, when a catastrophic event creates many DNA breaks, some will undergo breakage–replication–fusion while others will undergo breakage–fusion–replication; we refer to the latter as the breakage–replication/fusion cycle.

### Adjacent parallel breakpoints from DNA end replication

We first sought experimental evidence that a single DNA end can generate two adjacent parallel breakpoints. We exploited L1 retrotransposition to simultaneously generate and mark DSB ends. As described in a separate paper[43], transient L1 expression in p53-null RPE-1 cells generated both L1 insertions and translocation junctions containing reverse-transcribed L1 sequences. Both outcomes originate from DSB ends generated by the L1 open reading frame 2 protein (ORF2p), and are identified by the insertion of reverse-transcribed sequences (the 'primary' end of retrotransposition) and/or the presence of ORF2p endonuclease target sequences near the break site.

We identified multiple instances of adjacent parallel breakpoints in clones generated after L1 induction that had features indicating an origin from ORF2p-induced DSBs (Supplementary Note, Section 6). In the example shown in Fig. 1d, two nested deletions, each containing a truncated L1 insertion, indicate two pairs of adjacent parallel breakpoints (Fig. 1b). The sequence features at the two closest breakpoints (red and blue circles) directly relate them to L1 ORF2p, and the distances between each pair of parallel breakpoints (429 bp and 2,059 bp) are consistent with DSB resection[33–35]. Together, these observations demonstrate that breakage–replication–fusion can generate two parallel breakpoints from a single DSB end.

### Footprints of DNA end replication in human disease genomes

We next sought evidence of DSB end replication in human disease genomes. Although we cannot directly relate a rearrangement breakpoint to an ancestral DNA end, we can identify an ancestral DSB from breakpoints derived from reciprocal DSB ends: in particular, a reciprocal pair of parallel breakpoints directly identifies reciprocal DSB ends that undergo breakage–replication–fusion (Fig. 1b,d).

Based on the observations from the L1 clones (Fig. 1d and Supplementary Note, Section 6), we selected a heuristic threshold distance of 20 kb for the identification of adjacent parallel breakpoints (Methods). From 592,176 breakpoints detected in 2,588 cancers by the Pan-Cancer Analyses of Whole Genomes (PCAWG) study[16], we identified 20,795 pairs of adjacent parallel breakpoints from 1,793 samples. These breakpoints were identified at 35,422 rearrangement junctions (12% of all junctions), including 7,393 foldback junctions. Thus, adjacent parallel breakpoints are a widespread feature in cancer genomes.

For 3,138 pairs of adjacent parallel breakpoints, we identified one or multiple reciprocal breakpoints that demonstrate their origin from ancestral DSBs. There were 417 instances of reciprocal parallel breakpoints as shown in Fig. 1b. Among these were 53 instances of nested deletions (Supplementary Table 1) and 23 instances of reciprocal foldbacks (Supplementary Table 2), with examples shown in Fig. 1e. In the remaining instances, one or multiple breakpoints formed long-range translocations. Examples of reciprocal foldbacks were previously noted in ovarian cancers (supplementary fig. 8f of a previous publication[44]) but were assumed to result from BFB cycles. We suggest that these events arise from reciprocal DSB ends undergoing breakage–replication–fusion.

We further assessed the probability that adjacent parallel breakpoints were generated independently based on the distance between these breakpoints and their distance to the nearest breakpoint on the opposite side (Methods). This analysis showed that for 16,132 of 20,795 pairs of adjacent parallel breakpoints, the probability that they were generated independently was less than 5%.

In summary, adjacent parallel breakpoints are common in cancer genomes, and our analysis suggests that many of them are derived from sister DNA ends generated by breakage–replication–fusion.

### DNA duplication and amplification from breakage–replication/fusion

When the sister DNA ends are joined together in a single rearranged chromosome, this rearranged chromosome will contain a duplication

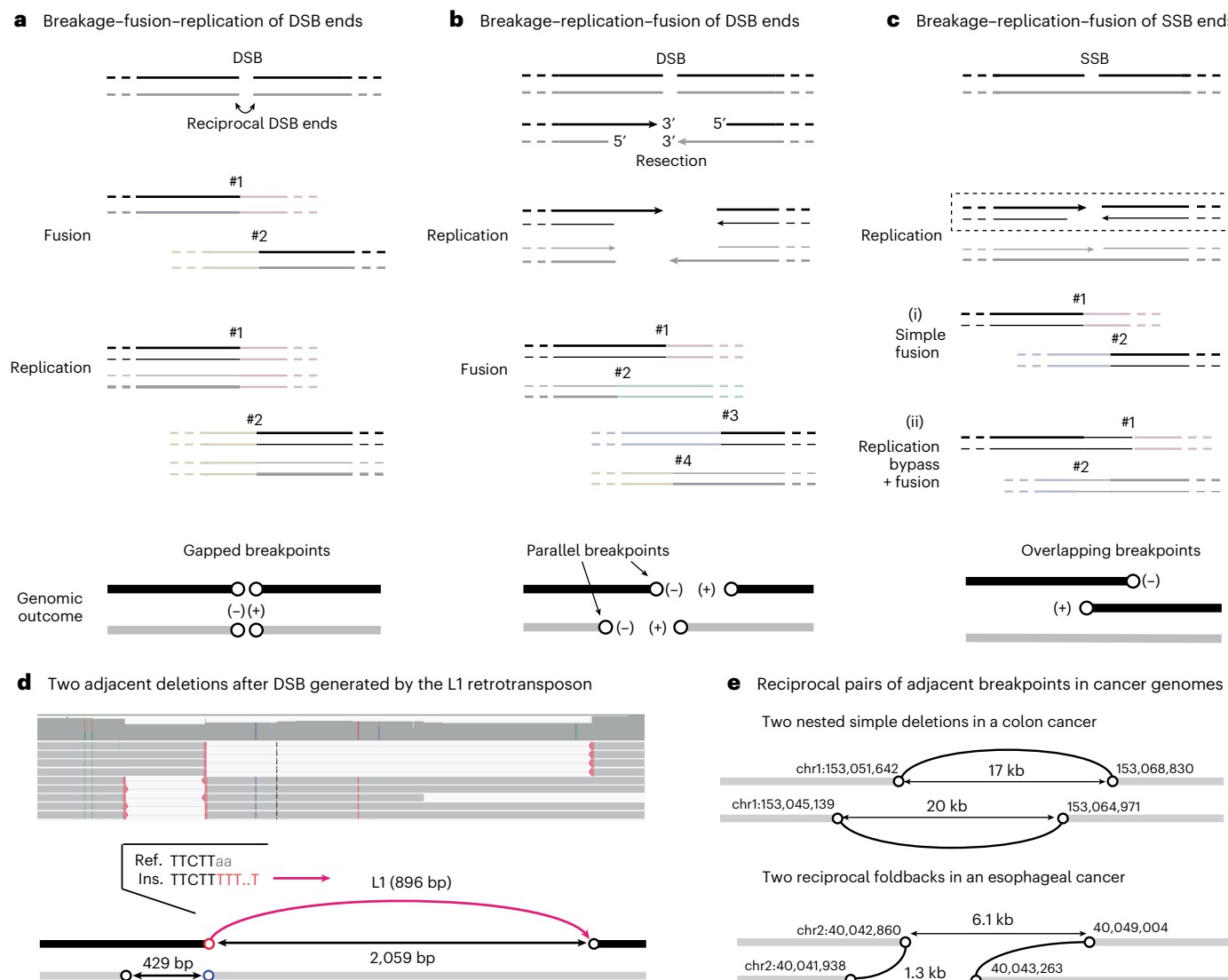

**Fig. 1 | Breakage–replication/fusion of DNA DSB and single-strand break ends.** In **a**–**c**, the two strands of the ancestral DNA are shown in black and gray: thick lines represent template DNA strands, thin lines represent newly synthesized DNA strands and arrows represent 3′ ends. Light-colored lines represent distal DNA sequences that are ligated to DSB ends derived from the original DNA. In the genomic outcomes, rearranged DNA is colored (black or gray) according to the ancestral DNA strand, and (−) and (+) denote the orientation of breakpoints defined by the directionality of copy-number transitions from left to right. Examples shown in **d** and **e** demonstrate the breakage–replication–fusion mechanism as shown in **b**. **a**, Breakage–fusion–replication of DSB ends. A single DSB generates two reciprocal DSB ends; each end is fused to a distal DNA end before replication (#1, #2), creating two reciprocal breakpoints separated by a small gap (open circles), termed adjacent gapped breakpoints. Note that after replication, both sister chromatids (black and gray) have the same breakpoints. **b**, Breakage–replication–fusion of DSB ends. Two DSB ends created by a single DSB undergo resection and replication, creating two pairs of replicated (sister) DNA ends that may undergo further end processing (not shown). Fusions of these DNA ends to distal DNA ends create four breakpoints in the rearranged sequences, one from each ancestral ssDNA end (5′ or 3′). Breakpoints derived from the 5′ end and 3′ end of a single ancestral DSB end (for example, #1 and #2) are adjacent and have the same orientation, termed adjacent parallel breakpoints. **c**, Breakage–replication–fusion of single-strand break (SSB) ends. Two ssDNA ends on the black strand are converted to two DSB ends by replication (dashed box). (i), The two DSB ends undergo simple fusions to create two gapped breakpoints, as in **a**. (ii), The DSB ends initiate homologous recombination using the intact sister chromatid (gray), creating two breakpoints with a small overlap by over-replication

(replication bypass); we refer to these as adjacent overlapping breakpoints. The example here also shows a sister-chromatid exchange (breakpoint #2 is now on the gray chromatid). See Extended Data Fig. 8c for additional information. **d**, Two adjacent deletions resulting from a single DSB owing to L1 retrotransposition in an experimentally generated clone of RPE-1 cells. These are the only L1 insertions identified on this chromosome (chr14) in this clone. The deletions are supported by long reads (top) and define two pairs of adjacent breakpoints (bottom). Each deletion junction contains a truncated L1 insertion and joins two DNA ends derived from a single ancestral strand (black or gray); the polarity of each ancestral strand is determined from the directionality of the reverse-transcribed L1 (complementary to the L1 messenger RNA, magenta arrows). The red circle (chr14:50,762,606) marks the ancestral 3′ end that underwent target-primed reverse transcription: this is established by the poly-T sequence (in red in the insertion sequence junction (Ins.)) that marks the initiation of reverse transcription and by the ORF2p EN target sequence at the breakpoint (TTcTT|aa in the reference sequence (Ref.)). The blue circle below the red (chr14:50,762,603) is derived from the 3′ end on the opposite strand that also underwent reverse transcription initiated from an internal position of the L1 mRNA, showing no poly-A/T. The two distal breakpoints (black circles) are inferred to be derived from the resected 5′ ends. **e**, Two examples of reciprocal breakpoint pairs in cancer genomes identified from the rearrangement junctions from a previous publication[16]. Top: two nested simple deletions in a colon cancer that are similar to **d** but without insertions. Bottom: two reciprocal foldbacks (direct joining of parallel breakpoints from each side) in an esophageal cancer. Note that the ancestral strand of each rearranged DNA segment cannot be definitively determined solely based on the breakpoints, as some DSB ends may have 5′ overhangs. Therefore, the rearranged DNA segments are all shown in gray.

**a** Duplication from one breakage–replication–fusion cycle

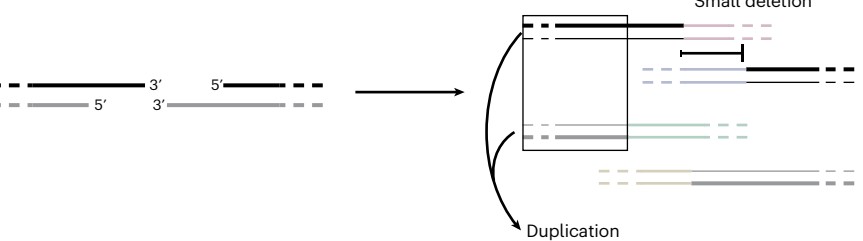

**b** Amplification through multiple cycles of breakage–replication/fusion

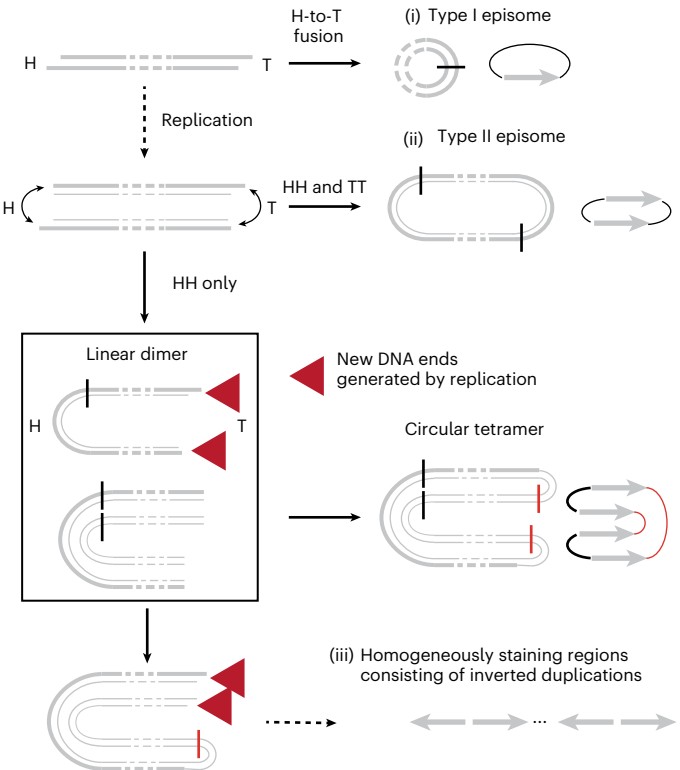

**c** Focal amplification at the *ERBB2* locus in a breast cancer

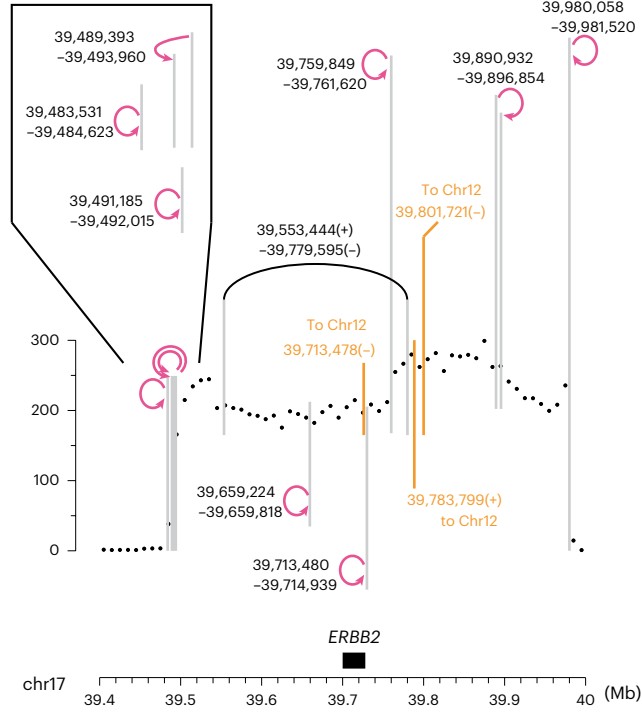

**Fig. 2 | Segmental DNA duplication and amplification from breakage–replication/fusion. a**, A single breakage–replication–fusion cycle can lead to DNA duplications when two sister DNA fragments are retained in a single rearranged chromosome and segregated into one cell. See Extended Data Fig. 1 for examples from human disease genomes. **b**, Three processes of amplification from an acentric DNA fragment. (i), A head (H)-to-tail (T) junction (black vertical line) joining opposite ends of the DNA fragment creates a type I episome. (ii), Fusions between sister DNA ends on opposite sides of replicated DNA create a type II episome. In both scenarios, the 'episome' (acentric extra-chromosomal circles) can be amplified by uneven segregation. (iii), If sister-end fusion occurs on one side (head-to-head, black vertical line) of replicated DNA, and sister ends on the opposite side remain unligated (red arrows), the outcome is a double-sized linear DNA fragment. Iterations of the same process can create a large array of amplified DNA with only head-to-head and tail-to-tail junctions (later fusion junctions shown as red vertical lines). The amplified DNA can be either circular or linear and consists of only inverted duplications. In the schematic diagrams

of amplified DNA on the right, the original DNA sequence is shown as a gray arrow to highlight the relative orientation of duplicated DNA. **c**, Amplification of the *ERBB2* oncogene in the HCC1954 genome that is consistent with linear DNA amplification as shown in **b**. The copy-number plot shows total sequence coverage in 10 kb bins. Breakpoints forming long-range rearrangement junctions are shown as vertical lines (three breakpoints joining chr12 are shown in orange); curved arrows represent foldback junctions between adjacent parallel breakpoints (positions labeled next to the curved arrows). See Supplementary Note, Section 7 for the copy-number data and rearrangement junctions of the entire chr17. Consider the three foldback junctions near 39.5 Mb within the 0.5 Mb amplicon: if they were generated by BFB cycles, the location of each foldback junction would correspond to the break site of a different dicentric chromosome bridge; the probability of generating two additional breaks within 10 kb from the first break is $(10 \text{ kb}/0.5 \text{ Mb})^2 = 0.0004$. Also note the proximity between the breakpoint at chr17:39,713,478 and the foldback junction between chr17:39,713,480 and 39,714,939.

(Fig. 2a). Moreover, the duplicated segments will be bounded by adjacent parallel breakpoints derived from sister DNA ends. Consistent with this prediction, we identified examples of copy-number gains flanked by adjacent parallel breakpoints in both human cancers and congenital diseases[14,15,45,46] (Extended Data Fig. 1 and Supplementary Note, Section 7).

Foldback junctions are the simplest outcome when sister DNA ends are joined together. We envision two processes by which a

double-stranded DNA (dsDNA) fragment can generate amplification with only foldback junctions (Fig. 2b). If both ends of a dsDNA fragment undergo breakage–replication–fusion to form foldback junctions (Fig. 2b (ii)), the outcome is a dimeric circular DNA (previously termed type II episomes[47]). Like simple monomeric DNA circles (type I episomes[47]; Fig. 2b (i)), dimeric DNA circles can fuel DNA amplification by asymmetric segregation over successive generations. This model

explains the amplification at the *AR* locus flanked by foldback junctions in a castration-resistant prostate cancer[46] (Extended Data Fig. 1a, right). Amplification can also occur on a linear acentric DNA fragment when the DNA ends on opposite sides fuse asynchronously (Fig. 2b (iii)). If sister DNA ends on one side are fused together, but sister DNA ends on the opposite side remain unligated (red arrows), the product is a linear inverted dimer. In the next cell cycle, another round of replication–fusion can create a circular or linear tetramer without any new breakage. Iterations of this process will produce a large tandem array of amplified DNA with 'nested' foldbacks that form homogeneously staining regions of inverted duplications[48,49].

One such example is the amplification spanning the *ERBB2* oncogene in the HCC1954 breast cancer genome (Fig. 2c). Similar patterns were also found in chr8p, chr12p and chr20q in this genome (Extended Data Fig. 2a–c and Supplementary Tables 3 and 4). Here, amplified *ERBB2* is contained in homogeneously staining regions[37,50] and is bounded by multiple foldback junctions previously attributed to BFB cycles[37]. However, the probability of generating foldback junctions in such close proximity by successive BFB cycles is very small (see Fig. 2c caption). Under the breakage–replication/fusion model, the close proximity between foldback junctions near 39.5 Mb is a natural consequence of the close proximity between the 3′ and 5′ ends of an ancestral DSB end (Fig. 2b (iii)). Moreover, if amplification takes place on a linear, extra-chromosomal DNA fragment, secondary breakpoints (both foldbacks and long-range breakpoints) can only arise within the amplicon, thus explaining the concentration of breakpoints within the amplified region (39.5–40 Mb). Importantly, in linear DNA amplification, amplified DNA is automatically doubled and linked in one chromosome that is segregated into one daughter cell, thus providing a more rapid route to higher DNA copy number than amplification by random segregation of episomal circles. The amplification of DNA copy number also does not require selection during the intermediate steps of amplification; therefore, focal amplifications lacking oncogenes (Extended Data Fig. 2a–c) may be passengers that undergo clonal fixation.

In summary, the presence of duplicated or amplified DNA segments flanked by adjacent parallel breakpoints suggests an origin from breakage–replication/fusion. From a single acentric DNA fragment, breakage–replication/fusion can generate dimeric DNA circles or a linear array of inverted duplications with closely spaced foldbacks, explaining the long-standing observation of inverted duplications in amplified DNA[47–49] that are unlikely to arise by multi-generational BFB cycles[37,38,44].

### Segmental copy-number gains after chromosome fragmentation

Above, we described the rearrangement and copy-number outcomes of breakage–replication/fusion occurring at a single dsDNA end and a single dsDNA fragment with two ends. Below, we describe the copy-number and rearrangement outcomes of breakage–replication/fusion after chromosome fragmentation.

We focused the analysis on an experimental model of chromothripsis (Fig. 3a, left) because this system enabled us to determine the structure of rearranged chromosomes with near-complete resolution (Methods). In a previous study[21], we used CRISPR–Cas9 to generate chromosome bridges containing dicentric chr4 and derived single cells with a broken chr4 (Supplementary Note, Section 8). In one generation, bridge breakage produced daughter cells with reciprocal DNA retention and deletion[21] similar to what was observed immediately after micronucleation[20]. However, over many generations, clones derived from single cells frequently had subclonal copy-number gains without reciprocal loss in the sibling clone[21]. The presence of copy-number gains in clones expanded after chromosome fragmentation was also observed in clones expanded after telomere crisis[24] (Supplementary Note, Section 9) or micronucleation[25] (Extended Data Fig. 3 and Supplementary Table 5).

One bridge clone (primary clone 1a from a previous publication[21], hereafter referred to as clone a) is interesting because the bulk DNA copy number oscillates between variable non-integer states that indicate subclonal copy-number variation (Fig. 3a, middle). Moreover, some subclones showed two-state copy number oscillation while others showed segmental copy-number gains (Fig. 3a, right, and Fig. 3b; also see fig. S22 from previous work[21]). The presence of subclonal copy-number variation enabled us to first determine the breakpoints of duplicated segments and then infer the evolutionary history of the rearrangements that produced the duplications (Methods and Supplementary Note, Sections 10 and 11). Based on a joint analysis of segmental DNA copy number (Supplementary Table 8) and rearrangement junctions (Supplementary Table 9), we determined both the structure (Extended Data Fig. 4) and the joining pattern (Extended Data Fig. 5) of nearly all duplicated segments in the subclones of clone a. In total, we identified 86 rearranged segments with sizes above 10 kb (Supplementary Tables 10–12) and 126 short insertions (<10 kb) between these segments (Supplementary Tables 13–16). We next show that the genomic features of the large segments, the short insertions and their arrangement in the rearranged chromosomes indicate that they all originate from breakage–replication/fusion of a single chromatid.

### Large duplications from breakage–replication/fusion

The origin of large duplications in clone a from ancestral chromosome fragments that underwent breakage–replication/fusion is established by two orthogonal lines of evidence that relate the boundaries of the duplications to ancestral DNA ends.

First, we identified 18 pairs of duplicated segments that are flanked by identical ('flush') or adjacent parallel ('staggered') breakpoints within 20 kb (Fig. 4). Knowing the exact size of each duplicated segment, we could assess the probability that the staggered breakpoints were generated independently using the ratio of breakpoint distance to the segmental size (Extended Data Fig. 6). Based on this metric, we determined that for 15 out of 20 pairs of staggered breakpoints, the probability of independent breakpoint generation was less than 0.05 (Supplementary Table 11). For the remaining five pairs, the breakpoint distances were within a similar range but the segments were shorter; therefore, all staggered breakpoints are consistent with an origin from the replication of (hyper)resected DSB ends. For three pairs of segments (Bb1/Bb2, Cb1/Cb2, Cc1/Cc2), the presence of reciprocal breakpoints directly established their origin from chromosome fragmentation (Extended Data Fig. 7a). These data provide statistical evidence that the staggered boundaries of duplicated segments arose from breakage–replication–fusion.

Second, we observed strand-coordinated base substitutions near the staggered breakpoints that directly established their origin from staggered DSB ends. Based on the breakage–replication/fusion model, the shorter breakpoint derives from an ancestral 5′ end and the longer breakpoint derives from an ancestral 3′ end. Thus, the offset region between the two breakpoints originates from the ancestral ssDNA overhang. We identified seven clusters of substitutions near the staggered breakpoints (Fig. 4, downward or upward arrows), six of which were restricted to the offset region (the only exception near the right side of the shorter Bb2 segment is explained in Extended Data Fig. 7b.) All the substitutions reflect deamination in the TpC context that is consistent with the outcome of ssDNA deamination by APOBEC enzymes[51]. Importantly, the signature of substitutions (C > X on the right side of each segment, downward arrows; G > X on the left side of each segment, upward arrows) directly established the deaminated ssDNA to be a 3′ overhang. Thus, the pattern of deamination between staggered breakpoints provides molecular evidence for their origin from staggered DSB ends. Additional evidence linking staggered breakpoints to staggered DSB ends comes from the coordination between breakpoints on opposite sides of duplicated segments (Extended Data Fig. 7c and caption).

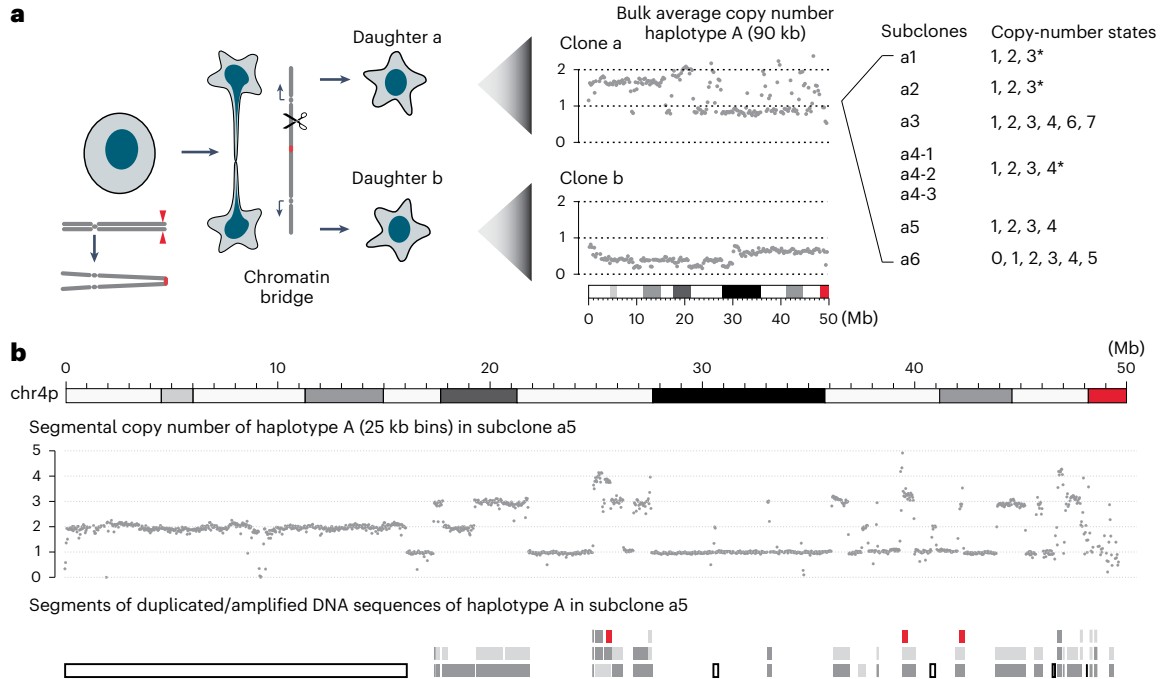

**Fig. 3 | Segmental copy-number gains in a clone expanded after chromosome fragmentation. a**, Left: experimental workflow; middle: bulk average DNA copy number of the 4A homolog (90 kb bins) in two clones, each derived from a daughter cell after breakage of a dicentric chr4 bridge. Note the non-integer copy-number states that indicate subclonal copy-number heterogeneity in both clones. Right: copy-number states in eight representative single-cell subclones derived from the top clone (clone a). Subclones a1 and a2 show mostly two-state copy-number oscillation (only one segment at three copies, indicated by ∗); a4 shows mostly three-state copy-number oscillation (only one segment at four copies, indicated by *); a5 shows four-state copy-number oscillation; a3 and a6 contain additional amplifications inferred to have been generated by secondary events. See Supplementary Table 8 for the complete segmental copy-number data of all the subclones. **b**, The copy number (25 kb bins, 4A haplotype) and rearranged segments of chr4p in subclone a5. There is an intact 4p copy in addition to the rearranged segments. Single-copy segments are shown as open bars, duplicated segments inferred to have been derived from sister DNA fragments by breakage–replication/fusion are shown as dark and light gray bars, and triplicated segments are shown as red bars. See Extended Data Fig. 5 for the order of rearranged segments in the rearranged chromosome.

Based on adjacent parallel breakpoints, we determined that 40 duplicated segments in clone a were derived from ancestral sister DNA fragments generated by breakage–replication/fusion (Supplementary Table 10).

### DNA over-replication from breakage–replication/fusion

In addition to nearly identical duplications generated by normal, semi-conservative replication of ancestral chromosome fragments, we also identified rare examples reflecting two mechanisms of DNA over-replication. The replication bypass mechanism[41,42] (Fig. 1c) explains two instances of overlapping duplications[18,52] (Extended Data Fig. 8a–c and caption); the second mechanism, leading to re-replication of a previously replicated DNA fragment, occurs when the previously replicated segment is fused to an unreplicated segment with unfired origins (Extended Data Fig. 8d and caption).

### Short insertions from breakage–replication/fusion

We identified 126 short insertions (median size, 184 bp) at the junctions between large duplications in clone a (Supplementary Tables 13–16). Three pieces of evidence indicate that both the insertions and the insertion rearrangement junctions are generated by chromosome breakage–replication/fusion.

First, when mapped to their origin sites, the insertions displayed several features indicating DNA fragmentation. Nearly all insertions (113 out of 126) were mapped to sites in close proximity (<10 kb) to breakpoints inferred to have been derived from ancestral DNA ends. Moreover, at several sites, the insertions lined up one after another in a tiling pattern, with little gap or overlap (Fig. 5a,b). The tiling pattern of insertions at the origin sites is incompatible with random

polymerase template-switching events in MMBIR that are expected to generate duplicated sequences with either large gaps or large overlaps at their original sites (Supplementary Video 4). Finally, seven tiles of insertions were mapped right next to breakpoints derived from the 5′ ends of ancestral DSBs (Figs. 4 and 5b and Supplementary Table 13). Similar patterns were also observed in other experimentally generated clones with chromothripsis (Supplementary Note, Sections 6, 9 and 14), in cancer genomes (Extended Data Fig. 9 and Supplementary Note, Section 7) and in congenital disorders[53]. Based on these observations, we suggest that many insertions originate as ssDNA fragments complementary to the 3′ overhang of resected DSBs. Two potential models for the generation of these insertions are discussed in Supplementary Note, Section 3.

Second, the joining pattern of insertions in rearranged DNA suggested DNA end-joining repair. A total of 111 out of 126 insertions were assembled into 17 chains (c1–c17) of two or more tandem insertions at rearrangement junctions (Supplementary Table 15), 13 of which are shown in Fig. 5c. These chains were only identified at junctions inferred to be breakage–replication–fusion junctions formed in S/G2, but not breakage–fusion–replication junctions formed in G1. Moreover, the junctions between the neighbor insertions within each chain often showed either >2 bp microhomology or additions of non-templated nucleotides. By contrast, breakage–fusion–replication junctions had few insertions and little microhomology, consistent with c-NHEJ in G1. Therefore, the insertion junctions probably arise from microhomology-mediated end-joining of sister DNA ends in breakage–replication–fusion.

Finally, and most definitively, the strand orientation of insertions at their destination junctions suggests that they were incorporated

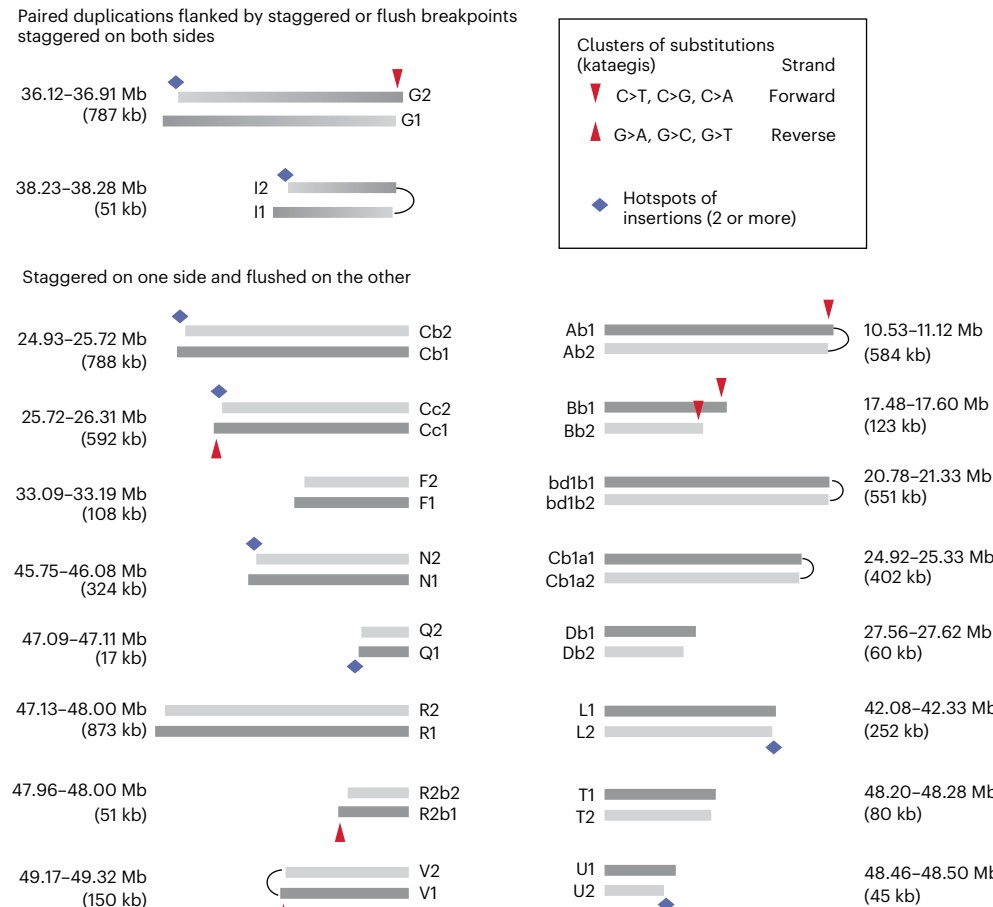

**Fig. 4 | Sister duplications in clone a defined by adjacent parallel breakpoints.** Each bar represents a rearranged segment (also see Extended Data Fig. 4); the coordinates of segmental breakpoints are listed in Supplementary Table 10. Arcs connecting adjacent breakpoints represent foldback junctions. Segmental sizes are labeled, but segments are not shown true to scale. Top: two pairs of duplications with staggered breakpoints on both sides. Dark and lighter ends correspond to boundaries inferred to be derived from ancestral 3′-ssDNA and 5′-ssDNA ends. Bottom: 16 pairs of duplications with flush breakpoints on one side and staggered breakpoints on the other side. Fusions between these segments create compound sister segments as shown in Extended Data Fig. 7c. Segments in darker gray are inferred to be derived from the ancestral DNA strands with a 3′ overhang. Red arrows point to regions with clustered substitutions (kataegis) indicating strand-specific cytosine deamination: downward arrows indicate deamination of cytosines on forward strand DNA (TpC>TpT, TpG or TpA); upward arrows indicate deamination of cytosines on reverse strand DNA (GpA>ApA and so on). Except for the kataegis cluster on the right end of segment Bb2 (explained in Extended Data Fig. 7b), all the other clusters are restricted to the offset region inferred to be the 3′ overhang of the ancestral DNA (dark gray) and show deamination signatures consistent with the DNA strands predicted by the breakpoints. Diamonds indicate regions corresponding to origins of multiple insertions (see Fig. 5).

into both DNA strands and could not have arisen from a conservative replicative process[14,16] such as MMBIR. Under the MMBIR model[11,14], insertions at each junction are continuously added to the 3′ end of the nascent leading strand that jumps from one template to the next; therefore, all the insertions would be added to a single strand in the rearranged DNA. As the original DNA strands of the insertions could be inferred from the adjacency between the insertions and nearby breakpoints (left-facing or right-facing arrows Fig. 5b), we were able to directly test whether the insertions were added to the same strand in the rearranged DNA. If we consider every pair of insertions that are next to each other in every insertion chain (Supplementary Table 15), 38 pairs are added to the same strand (arrows pointing to the same direction in Fig. 5c) but 41 pairs are added to opposite strands (arrows pointing to opposite directions). This observation therefore excludes MMBIR as the mechanism for generating the insertion junctions.

In summary, the genomic features of short insertions in clone a indicate that both the inserted sequences and the insertion junctions were generated in the same breakage–replication/fusion cycle that produced large duplications.

## Genomic complexity from one breakage–replication/fusion cycle

Based on the general assumption that breakpoints in close proximity arise at approximately the same time[16], we inferred that all the breakpoints and junctions in the ancestral rearranged chr4 of clone a (Extended Data Fig. 5c) were generated in a single breakage–replication/fusion cycle. Moreover, except for the rare instances of over-replication (Extended Data Fig. 8), all the ancestral segments, including short insertions, could be traced to non-overlapping ssDNA fragments. Therefore, the ancestral rearranged chr4 of clone a was most likely derived from a single ancestral chromatid over one breakage–replication/fusion cycle.

## Breakage–replication/fusion explains genomic complexity

A single breakage–replication/fusion cycle can generate both segmental duplications flanked by adjacent parallel breakpoints and rearrangement junctions containing insertions originating from DSB ends (Fig. 6). To assess the contribution of breakage–replication/fusion to insertion rearrangements in cancer genomes, we analyzed insertions in the PCAWG data. We identified 85,684 potential insertions with a

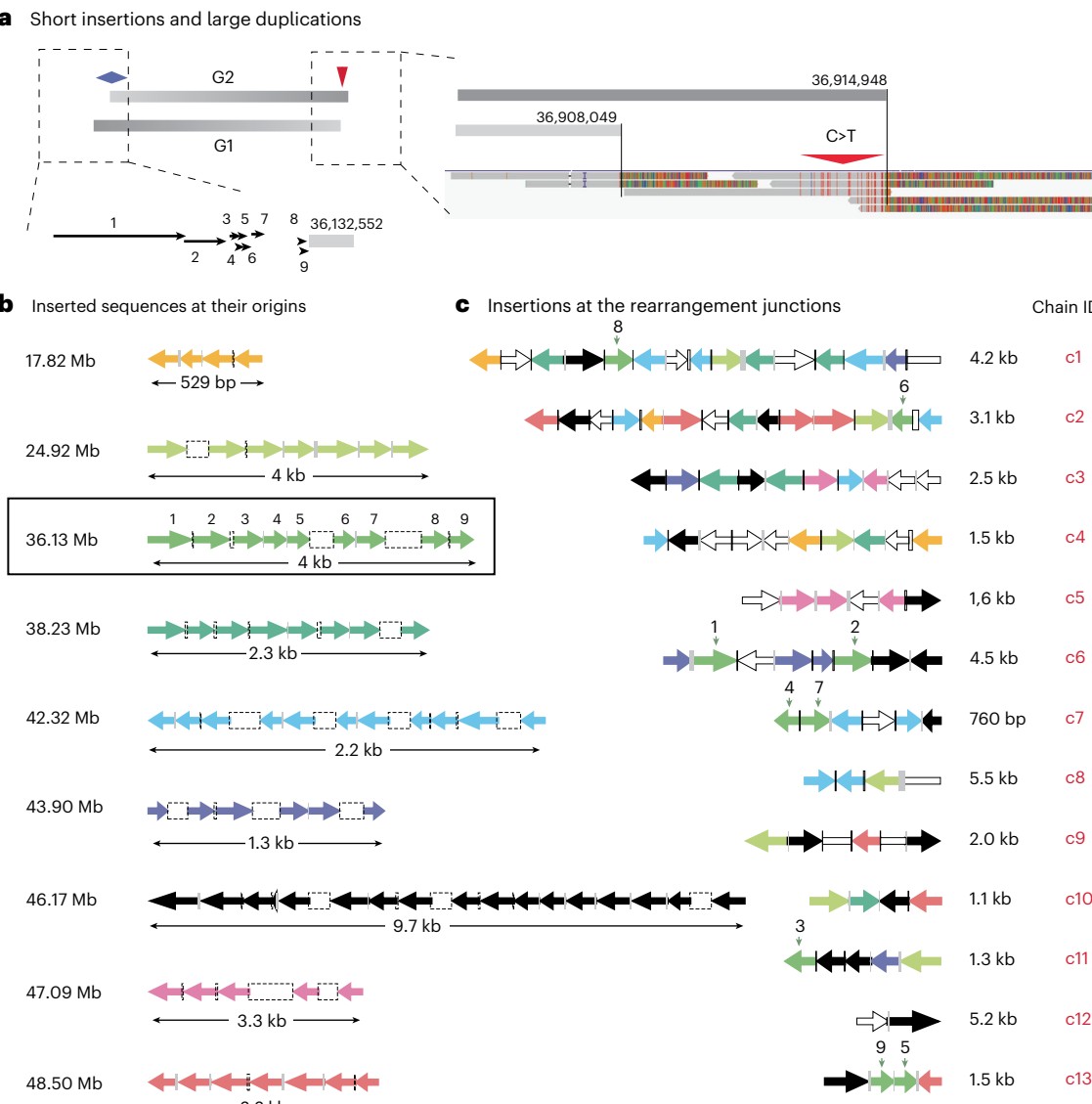

**Fig. 5 | Origin and arrangement of short insertions between large duplications in clone a. a,** An example of nine short insertions mapped to a region adjacent to the left breakpoint of the G2 segment shown in Fig. 4. The sizes and locations of the insertions (black arrows) are shown true to scale. We infer these insertions to have originated as ssDNA fragments of forward strand DNA based on the signature of deamination (C > T) on the opposite (right) end of the G2 segment. **b,** Tiling pattern of insertions at nine loci, including the example shown in **a** (36.13 Mb). Each tile consists of four or more short sequences that originate from adjacent locations but are identified at different destination junctions (shown in **c**). Insertions from each tile have the same color; the same color scheme is used in **c** to reflect the origin sites of each insertion. For example, the nine insertions mapped to 36.13 Mb are identified in junctions c1, c2, c6, c7, c11 and c13. See Supplementary Tables 13 and 15 for the mapping between the origins and destinations of all insertions. Both the size of each insertion (arrow) and the distance between neighbor insertions (open rectangles for gaps; filled rectangles for overlaps) are log transformed (same as in **c**). Except for the tile at 46.17 Mb, all the other tiles are adjacent to segmental breakpoints inferred to have been derived from ssDNA ends: the tile at 47.09 Mb is next to a breakpoint derived from an ancestral 3′ end; all the remaining tiles are next to breakpoints derived from ancestral 5′ ends. The original strands of insertions (left-facing arrows indicate ssDNA from the reverse strand; right-facing arrows indicate ssDNA from the forward strand) are inferred based on the strands of the ancestral DNA ends. **c,** Arrangement of insertions at 13 destination junctions (c1–c13) with two or more insertions ('chains' of insertions; see Extended Data Fig. 9a and Supplementary Table 15). Except for c13, which is assembled from short reads, all the remaining are resolved by both short and long reads. The color of each insertion reflects its origin, as shown in **b**; open arrows represent insertions from other regions. The directionality of each arrow indicates the strand of the inserted sequence in the rearrangement junction. Open bars without arrowheads (at junctions c1, c8 and c9) represent insertions whose original strands could not be determined. If a chain of insertions is generated by conservative DNA synthesis as in MMBIR, then all the inserted sequences have to be added to one strand; that is, the arrows need to point in the same direction. Clear violation of such strand coordination is seen in all chains except c11 and c12.

median size of ~2 kb (Methods and Extended Data Fig. 10a). These insertions accounted for 29% of all rearrangement breakpoints; 48% of insertions (41,445 out of 85,684) were mapped to regions within 10 kb from another breakpoint, but overlapping breakpoints were rare (<5% of insertions show 10 bp or larger overlap). These observations were consistent with the features of insertions generated by the breakage–replication/fusion mechanism (Fig. 5). Moreover, the two signatures of breakage–replication/fusion—adjacent parallel breakpoints and short insertions from a single DSB end—provide intuitive explanations for many complex rearrangement footprints that were identified in the PCAWG study[16] but to date had no mechanistic interpretation (Fig. 6b and Extended Data Fig. 10).

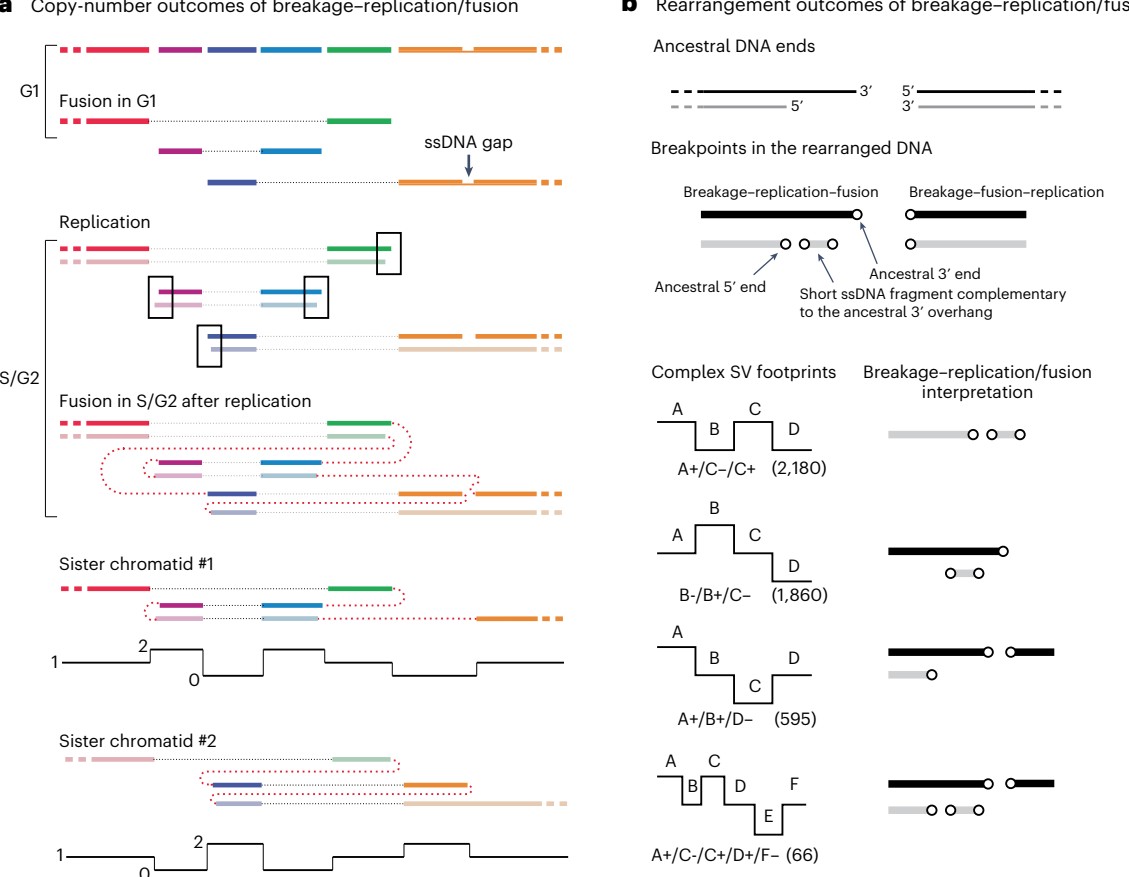

**Fig. 6 | Segmental copy-number alterations and rearrangement breakpoints generated by breakage–replication/fusion. a**, Segmental copy-number gain and loss generated by a breakage–replication/fusion cycle, including both breakage–fusion–replication and breakage–replication–fusion. The ancestral broken chromosome consists of six segments (shown in different colors) bounded by ten DSB ends; the rightmost segment also contains a single-strand gap with two ssDNA ends. Six DSB ends undergo ligation (fusion) in G1 (thin dotted lines), creating three chromosome fragments. After replication, there are seven new fragments (sister fragments are shown in dark and light colors) with ten new DSB ends: four pairs of sister DNA ends (outlined) plus two reciprocal DNA ends generated from the ssDNA gap. Fusions between the DSB ends in G2 (thick dotted lines) create reciprocal copy-number gains and losses on both sister chromatids. **b**, Footprints of rearrangement breakpoints generated by breakage–replication/fusion. Shown is one possible outcome when the left DNA end undergoes breakage–replication–fusion and the right DNA end undergoes breakage–fusion–replication. Top: ancestral DNA ends; middle: rearrangement breakpoints. Breakage–fusion–replication generates two flush breakpoints; breakage–replication–fusion generates two staggered breakpoints and a short insertion. Bottom: four complex structural variant (SV) footprints identified in cancer genomes[16] that can be explained by breakpoints generated in one breakage–replication/fusion cycle. Each footprint is represented by a collection of breakpoints on either the left (−) or the right (+) of adjacent segments (A, B, …). Note that the breakpoint orientation (+/−) in the original study[16] is opposite to our convention. The first three footprints (all having three breakpoints) were discussed in the supplementary information of the original study (pages 76–81); the last footprint with five breakpoints was shown in supplementary fig. 48 (page 82) of the same study. Numbers in parentheses represent the total counts of instances of each footprint reported in the original study, regardless of the joining pattern between breakpoints.

## Discussion

In this study, we describe the breakage–replication/fusion cycle, a rearrangement process that can generate DNA duplication and amplification after DNA breakage or chromosome fragmentation.

Our finding that one DNA end can generate two breakpoints by normal (S phase) DNA replication (Fig. 1b) revises the common assumption in genome evolution that each rearrangement breakpoint is derived from a unique DNA end[54]. This outcome of breakage–replication/fusion has some conceptual similarity to the generation of different substitutions from a single DNA base lesion[55]. However, unlike base substitutions, rearranged sequences generated by breakage–replication/fusion can be joined together to create DNA duplication within one cycle (Fig. 2a). Moreover, multiple cycles of breakage–replication/fusion can generate high-level DNA amplification (Fig. 2b and Extended Data Fig. 2d), including homogeneously staining regions consisting of inverted duplications in cancer genomes (Fig. 2c and amplified *BCR-ABL* fusion in K-562 cells[56]).

Chromosome breakage–replication/fusion can occur after chromosome shattering in micronuclei or bridges[20,21,24,25]. Damaged chromosomes from these structures contain a large number of dsDNA and ssDNA breaks[57,58]. In the first cell division, the distribution of chromosome fragments between daughter cells[20,21] explains the oscillating copy-number pattern in canonical chromothripsis[19]. However, after incorporation into the daughter nuclei, breakage–replication/fusion of chromosome fragments with unligated DNA ends can generate both deletions and duplications in the next cell cycle (Fig. 6a), explaining segmental copy-number gains observed after clonal expansion[24,25]. Therefore, one instance of chromosome fragmentation is sufficient to generate segmental gains that are commonly observed for chromothripsis in human cancers[19,23], without invoking aberrant DNA synthesis[14,16] or additional instances of DNA breakage[27,28]. Finally, for a chromosome that has been trapped in a micronucleus over multiple generations, we speculate that breakage–replication/fusion can produce amplified DNA with an 'onion-skin' structure that mimics the

original model of DNA amplification by re-replication[8] (Supplementary Note, Section 5).

Another genomic hallmark of breakage–replication/fusion is junctions with complex insertions, which were previously observed both after micronucleation[20] and after bridge breakage[21]. Insertion junctions are well-described features of complex rearrangements in both human disease genomes[14,16,19,22,23,37,59] and experimental models[60–62], and they are frequently attributed to DNA polymerase template-switching events[11,14,16]. Here, our analyses of insertion rearrangements in cancer genomes, experimental clones and single cells (Fig. 5, Extended Data Fig. 9, and Supplementary Note, Sections 7, 9 and 14) have revealed two common features of insertions—the tiling pattern at their origin sites and the random strand orientation at their destination sites—that are incompatible with the template-switching model.

We propose the following speculative model for the generation of complex insertion junctions (Supplementary Video 4; also see Supplementary Note, Section 3). (1) The tiling pattern of insertions (Fig. 5b) suggests that they originate from ssDNA generated by discontinuous gap-filling DNA synthesis[36,63]. (2) The presence of microhomology and strand alternation between neighboring insertions in the rearranged DNA (Fig. 5c) suggests that the ssDNA fragments are incorporated into both DNA strands, probably initiated by microhomology-mediated annealing[64] followed by fill-in synthesis and ligation. (3) For these fragments to be incorporated in the rearranged DNA, they must first be displaced from the original template. The displacement may involve a helicase such as the Bloom helicase, which was implicated in the conceptually similar process of C-circle formation during alternative lengthening of telomeres[65].

The breakage–replication/fusion mechanism has several implications for the interpretation of cancer genomic observations.

First, the generation of adjacent parallel breakpoints and short insertions in a single breakage–replication/fusion cycle can explain many previously described but unexplained patterns of complex rearrangements (Fig. 6b and Extended Data Fig. 10).

Second, a single breakage–replication/fusion cycle can generate duplications of three different sizes: insertions (≲1 kb; Fig. 5), over-replicated DNA (10–100 kb; Extended Data Fig. 8) and large duplications (≳1 Mb; Fig. 4). Appealingly, the three sizes correspond to the tri-modal size distribution of duplications in cancer genomes (with peaks at ~1 kb, ~10 kb and >100 kb)[16]. We suggest that duplications of each size range are generated by a different mechanism of DNA synthesis with differing processivities (gap-filling synthesis, replication bypass and semi-conservative replication), instead of a single process of aberrant replication that would generate 100 bp insertions in one circumstance and 1 Mb duplications in another[16].

Third, breakage–replication/fusion explains the paucity of partial overlap between duplicated sequences. This contrasts with the MMBIR model[14,16], which should generate duplicated segments with all possible degrees of partial overlap because of random strand invasion (Supplementary Video 4). Similarly, partial overlap between duplicated segments is also expected for random breakage of two sister chromatids[27] (Supplementary Note, Section 2). Notably, the rarity of partially overlapping segments is also observed in focal amplifications in the HCC1954 breast cancer cell line (Supplementary Note, Section 7) and in primary cancers (for example, Fig. 6e of a previous publication[17]). These observations suggest that breakage–replication/fusion, rather than the other two models, be the dominant mechanism for generating high-level DNA amplification.

Finally, breakage–replication/fusion can generate foldback junctions at DNA ends that are not restricted to the ends of broken bridge chromosomes. We suggest that many foldbacks in cancer genomes, especially those without complete deletion of sequences on the telomeric side of the breakpoint, are derived from breakage–replication/fusion, not BFB cycles[37,38].

In summary, the breakage–replication/fusion mechanism expands the repertoire of genomic outcomes derived from DNA breakage, substantially extends the model of chromothripsis and suggests a new, simplifying mechanism for high-level DNA amplification.

## Online content

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

## Methods

This study complies with all relevant local ethical regulations. Access to data controlled by the International Cancer Genome Consortium (ICGC) was granted by the ICGC data access compliance office (DACO-2877).

### Identification of adjacent breakpoints in published data

To identify adjacent parallel breakpoints in the ICGC's whole-genome cohort, we started with 296,088 curated structural variants identified in 2,588 samples by the PCAWG Structural Variation Working Group. This dataset was from an early release (20170720). The final dataset[16] contained 274,515 structural variants. As each structural variant (or junction) contains two breakpoints, there were 592,176 breakpoints in total. For breakpoints in each sample, we first identified adjacent (+/−) breakpoints; that is, those corresponding to insertions or adjacent overlapping breakpoints. We did not distinguish between breakpoints associated with insertions or overlapping segments (Extended Data Fig. 10a). Adjacency was defined by a breakpoint distance of less than 20 kb. From the remaining breakpoints, we then identified adjacent gapped breakpoints (−/+) and adjacent parallel breakpoints (+/+, −/−) based on the same threshold distance. The exclusion of breakpoints with insertion or overlapping adjacency (+/−) from gapped (−/+) or parallel (+/+, −/−) adjacency was necessary because the breakpoints of adjacent insertions (Fig. 5) all fall within the threshold but are generated by a different process from adjacent gapped or parallel breakpoints (Fig. 1). Two or more adjacent parallel breakpoints were grouped together if they were within 20 kb from each other and had the same orientation. Adjacent parallel breakpoints with reciprocal breakpoints were identified by intersecting the set of adjacent parallel breakpoints with the set of adjacent gapped breakpoints.

For adjacent parallel breakpoints, the likelihood that they were generated independently can be estimated by the ratio of the distance between the parallel breakpoints to the size of the ancestral segment from which the two parallel breakpoints were derived (Extended Data Fig. 6a). For cancer genomes, the size of the ancestral segment cannot be determined exactly. However, the ancestral segment has to be longer than the distance between the parallel breakpoints and the nearest breakpoint on the opposite side (not the reciprocal side): for +/+ breakpoints, the nearest breakpoint is the first (−) breakpoint to the right; for −/− breakpoints, the nearest breakpoint is the first (+) breakpoint to the left (again, insertion and overlapping breakpoints were excluded). We thus estimated breakpoint independence using the ratio of the distance between the adjacent parallel breakpoints to their distance from the nearest opposite breakpoint; this ratio represents an upper bound of the breakpoint distance ratio, as shown in Extended Data Fig. 6a.

Adjacent parallel breakpoints in prostate cancers[45,46] and in congenital genetic diseases[14,15] were identified manually from the figures and the supplementary tables and figures in the original publications.

### Generation of subclonal sequencing data of bridge clone a

The ancestor cell of bridge clone a had received a broken chr4 from a dicentric chr4 bridge. The dicentric chr4 was generated by transient Cas9 induction (1 μg ml$^{-1}$ doxycycline for 14–16 h) in retinal pigmental epithelial cells (RPE-1) that constitutively expressed sgRNA targeting a subtelomeric sequence (5′-TTTAGTGCCCGGCCGCAAGG-3′) present on both chr4 homologs. Bridge formation and breakage was identified by live-cell imaging[21]. Each daughter cell was expanded for approximately 4 weeks (>10$^6$ cells) under Rb and p21 suppression by short hairpin RNAs[21,24] to generate the primary bridge clones. Single cells from the primary clones were flow-sorted and expanded for approximately 2 weeks to generate subclones (10$^4$–10$^5$ cells) that were used for bulk DNA library construction. In the original study[21], 23 single-cell-derived subclones were generated. We selected nine subclones (including two showing the same copy-number of chr4A

represented as subclone a1) to perform deep whole-genome sequencing analysis. The sequencing data of these nine subclones were generated from new sequencing libraries constructed from the original cell pellets. The accession numbers of the new sequencing data are listed in Supplementary Table 7.

### Generation of whole-genome sequencing data of L1 clones

Clones expanded from single RPE-1 cells with transient L1 expression were generated in a previous study[43]. Both shotgun and long-read whole-genome sequencing data were generated for these clones; the experimental protocols have been previously described[43].

### Sequencing data processing

The DLD1 sequencing data from a previous publication[25] were aligned to the T2T-CHM13v2.0 assembly (https://www.ncbi.nlm.nih.gov/datasets/genome/GCF_009914755.1). All remaining sequencing data were aligned to the GRCh38 reference. All short-read data were aligned with bwa mem (v.0.7.13-r1126 for the RPE-1 data; v.0.7.17-r1188 for DLD1 clones) using default parameters and processed as described previously[56]. Long reads were aligned using minimap2 (v.2.7-r654, '-ax map-pb' for the bridge clone; v.2.26-r1175, '-a -k19 -w19 -U50,500 -g10k -A1 -B4 -O6,26 -E2,1 -s40' for the HCC1954 data). Alignment and post-alignment processing of the sequencing data of L1 RPE-1 clones have been previously described[43].

### Detection of single-nucleotide substitutions in bridge clone a

Detection of short variants (single-nucleotide substitutions and small insertions and deletions) was performed using GATK (v.4.2.4.1) HaplotypeCaller as described previously[21]. We manually reviewed clustered mutations near breakpoints that are reported in Supplementary Table 10. Both copy-number calculation and short variant discovery were performed using only short-read data.

### DNA copy number calculation

The total sequence coverage (10 kb intervals) was calculated from the number of reads passing filters as described previously[21,56]. DNA copy number for the DLD1 cells (Extended Data Fig. 3) was calculated from the normalized total sequence coverage. Haplotype phasing and haplotype-specific copy number of the HCC1954 genome (Fig. 2 and Extended Data Fig. 2) were calculated as described previously[56] and the results were reported in a separate study[50]. Haplotype phasing and haplotype-specific DNA copy number for all the RPE-1 samples were calculated using the RPE-1 haplotype data and the workflow as described previously[21,56].

### Detection and validation of rearrangement breakpoints and junctions

Rearrangement junctions were detected from both long reads and short reads using the same algorithm as described previously[21]. In brief, long reads with multiple split alignments were represented as discordant reads at each split junction. Rearrangement junctions were identified by clustering of discordant reads (both short reads and long reads). The manual review and curation of rearrangement junctions containing L1 insertions (Fig. 1d) was previously described[43]. For adjacent breakpoints (parallel, overlapping or insertion) in the HCC1954 genome (Supplementary Tables 3, 4, 20 and 21), we manually reviewed every event by either PacBio or Nanopore reads; for the three long insertion chains listed in Supplementary Tables 22–24, we listed the supporting reads for each junction. For the DLD1 clones, the breakpoints (Supplementary Table 6) were identified by manually reviewing all copy-number transitions on chrY in the following regions where short reads can be uniquely mapped: 2.4–5.9 Mb, 6.15–8.9 Mb, 9.9–11.7 Mb, 12.7–14.8 Mb, 15–17 Mb, 17.6–18.3 Mb and 19.5–22.5 Mb; if the partner breakpoints of breakpoints in these regions fell into repetitive regions within 0–27.5 Mb, we also listed the possible locations; if the

partner breakpoints were mapped to regions past 27.5 Mb, they were omitted. For rearrangement junctions in bridge clones or subclones (Supplementary Tables 9, 17 and 19), we manually reviewed each junction and verified breakpoints associated with copy-number transitions (Supplementary Table 8). We further listed names of supporting reads for each junction.

## Haplotype phasing of rearrangement breakpoints in RPE-1 clones

Rearrangement breakpoints in RPE-1 clones were assigned to the parental haplotype based on haplotype-specific copy-number transitions. For copy-number transitions without detectable rearrangement junctions, we manually reviewed the sequencing data to identify the breakpoints: nearly all of these breakpoints form junctions with repetitive sequences (for example, centromeric, telomeric or rDNA repeats). We aligned the junction sequences to the CHM13 reference to identify the most likely locations of these sequences (for example, chr13p) without the exact coordinates. Rearrangement breakpoints and junctions from both homologs of chr4p in all subclones of bridge clone a are listed in Supplementary Tables 9 and 17.

## Determination of rearranged segments in bridge clone a

Our ability to uncover the full spectrum of copy-number and rearrangement outcomes of breakage–replication/fusion following chromosome fragmentation depends on the determination of both the rearranged segments in bridge clone a and the ancestral DNA fragments that give rise to these segments. This analysis was enabled by three experimental and analytical advances: (1) the induction of DNA fragmentation on a bridge chromosome, which is more likely to generate a derivative chromosome with a functional centromere (in comparison to micronucleation) that increases its chance of preservation in the progeny clone; (2) the deep analysis of subclones to track down most of the ancestral DNA fragments including insertions; and (3) the joint analysis of subclonal copy-number and rearrangement variation to infer both the structure and the evolutionary history of rearrangements.

The central problem in the determination of rearrangement segments is to identify the two boundaries (breakpoints) of each segment (see Supplementary Note, Section 1). This is especially challenging for copy-number gains because of the multiple overlapping segments, for which there is ambiguity about the phasing of breakpoints on each duplicated segment (see Supplementary Note, Section 2). This problem was solved by the following strategies. First, we determined the copy number of each breakpoint based on the segmental copy-number difference across the breakpoint. For adjacent breakpoints for which the copy-number change at each breakpoint could not be determined directly, we used the copy number of the partner breakpoint to determine the copy number of each breakpoint. Second, if a region of copy-number gain or retention is flanked by deletions, the breakpoints on opposite sides of the retained segments directly determine the boundaries of the segment. This was applied to segments in the a6 subclone with interspersed deletions of the 4A homolog. Once the two breakpoints of a segment were determined in one subclone (for example, a6), we assumed that the same segment was retained in other subclones if the same breakpoints were also identified in the other subclones. This assumption implies that there is no homologous recombination between the duplicated sequences at different loci (that is, non-allelic homologous recombination). Third, in regions with variable or multi-copy gains, we assessed the copy number of each breakpoint in different subclones to identify breakpoints that maintained the same copy-number state, indicating their association with a single segment. Finally, if a region with multi-copy gains is flanked by adjacent parallel breakpoints on both sides, we imposed the constraint that there must be two segments generated from an ancestral dsDNA fragment by replication, such that the staggered ends of the ancestral dsDNA fragment became two pairs of adjacent

breakpoints. For details of this analysis, including the copy number and structure of all rearranged segments in each subclone, see Supplementary Note, Sections 10 and 11.

## Assembly of complex junctions with multiple insertions

We assembled insertions at rearrangement junctions by two approaches. When long-read data were available (bridge clone a and HCC1954), the insertions were identified directly from the split alignments of long reads and refined by split short reads. When long-read data were unavailable, we manually assembled complex junctions from short-read data as described previously[21]. For bridge clone a, the complex junctions were independently assembled from the short-read data and validated by long reads. As a large fraction of the insertions were mapped to a few hotspots (Fig. 5), we re-aligned the junction sequence (either assembled from short reads or directly from long reads) to these regions using minimap2 (v.2.7-r654) with different parameters '-k13 -H -n2 -m14 -s28 -w5 -g 35' and '-k17 -H -n2 -m20 -s40 -w5 -g 35' to increase the alignment sensitivity for short insertions. The original locations of the insertions and the order of insertions in the rearranged DNA are listed in Supplementary Tables 13–16.

## Statistics and reproducibility

Statistical analysis was performed only for breakpoint independence (Extended Data Fig. 6a), and the procedure is described in the main text. No other statistical test was performed. No statistical method was used to predetermine sample size.

## Reporting summary

Further information on research design is available in the Nature Portfolio Reporting Summary linked to this article.

## Data availability

Whole-genome sequencing data (both Illumina short reads and PacBio long reads) of L1 clones are available from the NCBI Sequence Read Archive (SRA) under BioProject PRJNA1197453. Long-read sequencing data of the HCC1954 genome (both PacBio and Nanopore) were downloaded from the NCBI SRA (BioProject PRJNA1086849). The Hi-C and short-read sequencing data of the HCC1954 and its matching germline reference HCC1954BL are available from the SRA under BioProject PRJNA1079784. Short-read sequencing data of DLD1 clones from a previous publication[25] were downloaded from the European Genome–Phenome Archive (EGA) under EGAD00001004163. Short-read sequencing data of post-crisis RPE-1 clones from previous work[24] were downloaded from the EGA under EGAD00001001629. All the sequencing data of bridge clones and subclones are available from SRA under PRJNA602546.

## Code availability

Scripts for the PCAWG breakpoint analysis and the complete results of all the genomic analyses are available at the GitHub online repository (https://github.com/chengzhongzhangDFCI/BRF).

## Acknowledgements

We thank J. Haber for suggesting the connection between adjacent parallel breakpoints and resected dsDNA ends, for discussions related to break-induced replication and for comments on the paper; H. Li for suggestions about long-read alignment and for discussions on the HCC1954 genome analysis; and N. Umbreit for discussions in the early stage of this study. The whole-genome sequencing libraries of bridge subclones were generated by J. Wang with help from N. Umbreit. The sequence coverage of DLD1 clones was generated by S. Turner. We thank the following people for suggestions and comments on the paper: E. Hodis, M. Leibowitz, M. Leventhal and T. Ubih. Funding for this study comes from the Claudia Adams Barr Program for Innovative Cancer Research (to C.-Z.Z.) and the Innovations Research Fund from

Dana-Farber Cancer Institute (to C.-Z.Z. and K.H.B.), the Ludwig Center at Harvard University (to D.P.), Alex's Lemonade Stand Foundation for Childhood Cancer (to C.-Z.Z.), BreakThroughCancer (to C.-Z.Z.), and NCI (R01CA276112 to K.H.B. and C.-Z.Z., R35CA293978 to D.P.). D.P. is supported by the Howard Hughes Medical Institute.

## Author contributions

C.-Z.Z. conceived the study and performed all the analyses. C.M.-D. and K.H.B. generated the L1 clones and the sequencing data shown in Fig. 1d and Supplementary Data. D.P. contributed to the interpretation and conclusions. C.-Z.Z. and D.P. wrote the paper with contributions from C.M.-D. and K.H.B.

## Competing interests

C.-Z.Z. is a co-founder and scientific adviser for Pillar BioSciences. D.P. is a member of the Volastra Therapeutics scientific advisory board. The other authors declare no competing interests.

## Additional information

**Extended data** is available for this paper at https://doi.org/10.1038/s41588-025-02434-5.

**Correspondence and requests for materials** should be addressed to Cheng-Zhong Zhang.

**a**   Adjacent parallel breakpoints flanking amplified *AR* in prostate cancers

**Fig.4A** from <u>Viswanathan, Ha et al. (2018)</u>

**Fig.3G** from <u>Zhou et al. (2022)</u>

breakpoints from **Table S5, tab B**

3.8kb ← 65765375
65761593

69619949 → 38kb
69658154

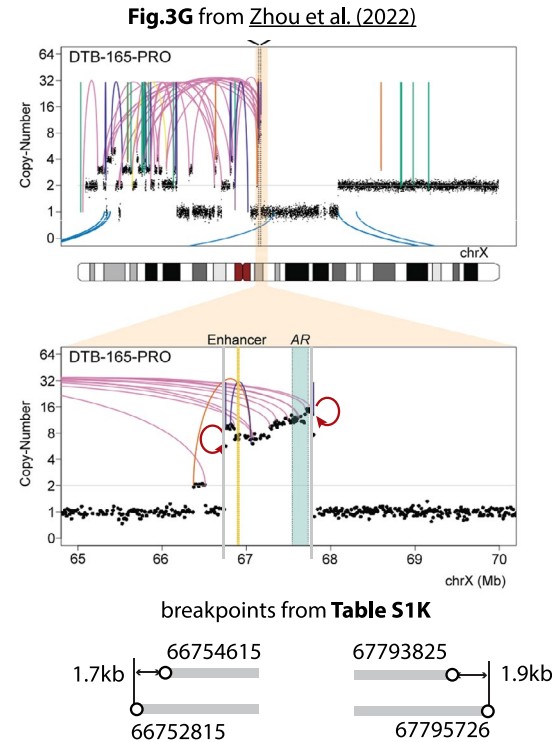

breakpoints from **Table S1K**

1.7kb ← 66754615
66752815

67793825 → 1.9kb
67795726

**b**   Adjacent parallel breakpoints flanking copy-number gains in congenital disorders

**Fig. 2A** from <u>Liu et al. (2011)</u>

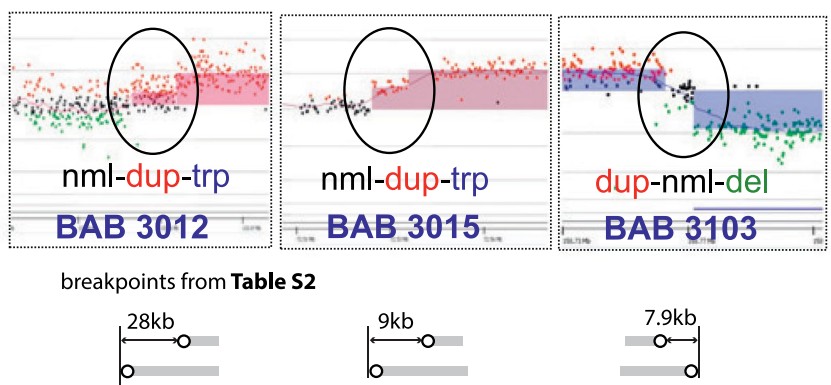

nml-dup-trp       nml-dup-trp       dup-nml-del
**BAB 3012**      **BAB 3015**      **BAB 3103**

breakpoints from **Table S2**

28kb              9kb               7.9kb

Two pairs of adjacent breakpoints flanking a triplication in case BAB3097 from <u>Liu et al. (2017)</u> (**Data S1**, pg. 4)

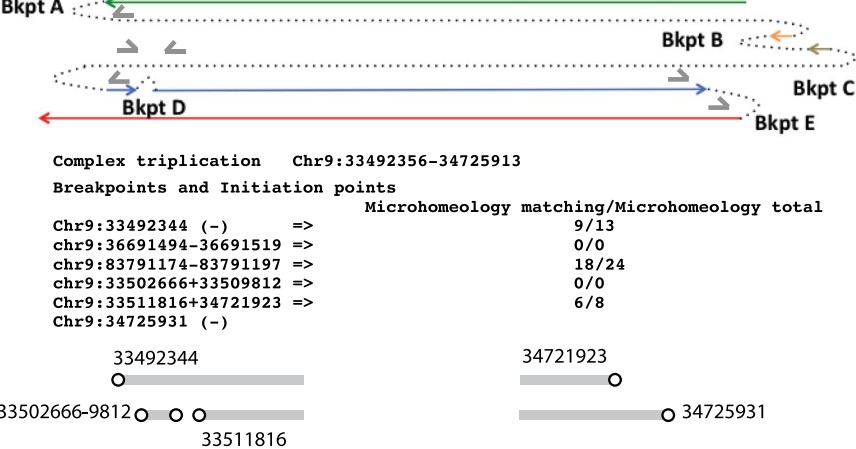

```
Complex triplication   Chr9:33492356-34725913
Breakpoints and Initiation points
                       Microhomeology matching/Microhomeology total
Chr9:33492344 (-)      =>            9/13
chr9:36691494-36691519 =>            0/0
chr9:83791174-83791197 =>            18/24
Chr9:33502666+33509812 =>            0/0
Chr9:33511816+34721923 =>            6/8
Chr9:34725931 (-)
```

33492344                                    34721923

33502666-9812                               34725931
33511816

**Extended Data Fig. 1 | See next page for caption.**

**Extended Data Fig. 1 | Adjacent parallel breakpoints identified in cancer and congenital disease patient samples from previously published studies.** All figure panels were from the original papers; please refer to the original publications for the complete caption (Refs. 45 and 46 for panel **a**; Refs. 14 and 15 for panel **b**). Adjacent breakpoints and insertions (all reported in the original papers) are annotated below each figure panel. **a.** Adjacent parallel breakpoints identified in two castration-resistant prostate cancers near the *AR* (Androgen Receptor) locus on chrX (green area for the *AR* gene; yellow line for the *AR* enhancer). Note that the left example is shown in GRCh37 coordinates, the right example is shown in GRCh38 coordinates. In both examples, the amplified DNA segment is flanked by adjacent parallel breakpoints on both sides (red arrows on the left figure, curved arrows representing foldback junctions on the right). In the left example, the distance between the two breakpoints near 69.6 Mb (38 kb) exceeds the 20 kb threshold; however, their origin from a single ancestral DSB end is supported by the probability of breakpoint independence estimated from the ratio of breakpoint distance (38 kb) to the segmental size (3 Mb) of the duplication (cf. Extended Data Fig. 6a). The example shown on the right was originally suggested to have been generated by breakage-fusion-bridge cycles after chromothripsis (note the oscillating pattern of copy-number gains on the Xp arm). We suggest that the amplification occurred through multiple breakage-replication/fusion cycles of a linear or circular acentric DNA fragment after chromothripsis (cf. Figure 2b and Extended Data Fig. 2d). **b.** Adjacent parallel breakpoints in congenital genetic disorders. The top panels show three instances (BABxxxx) of a small copy-number segment with an intermediate copy-number state between two flanking copy-number segments. The size of the intermediate copy-number segment (inferred from microarray data) corresponds to the distance between adjacent parallel breakpoints as indicated below. The bottom panel shows adjacent parallel breakpoints (determined from sequencing of the rearrangement junctions) flanking a triplication.

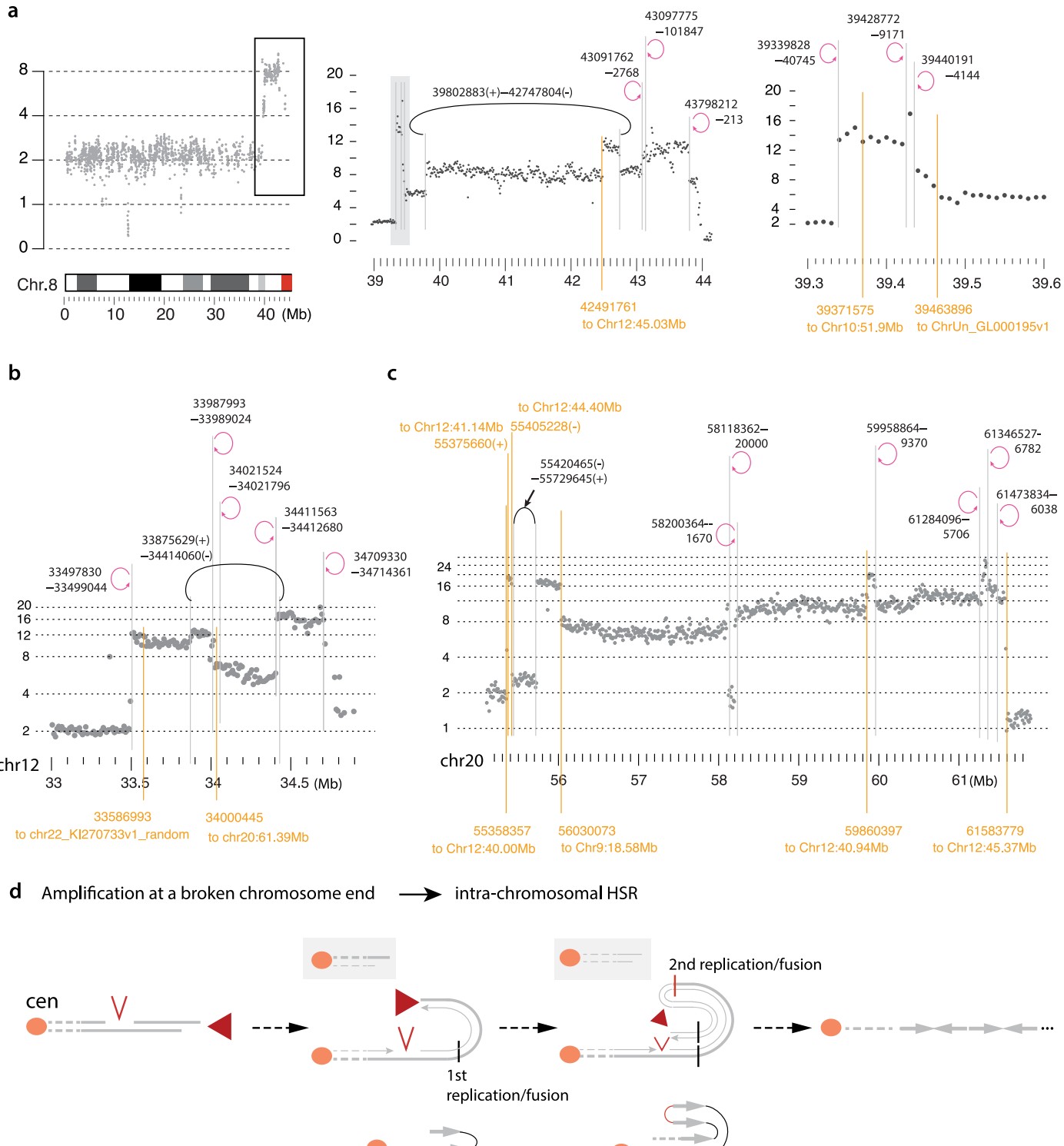

**Extended Data Fig. 2 | Additional examples of amplifications with nested foldbacks in HCC1954.** All copy-number plots are calculated from the total sequence coverage (10 kb bins). In **a**, the copy number is specific to one homolog as the other homolog is completely deleted; in **b**, the copy number is the combined copy number of both homologs, including two copies of the unamplified homolog; in **c**, the copy number is calculated by subtracting the copy number of the unamplified homolog from the total copy number. Breakpoints of intra-chromosomal junctions are marked with vertical lines; breakpoints of inter-chromosomal junctions are marked with orange lines; foldbacks are shown as magenta curved arrows. **a**. Amplification in a 4 Mb region (box) on 8p containing six foldbacks, three inter-chromosomal, and one intra-chromosomal junction (black line). The middle and right panels show zoomed view of the DNA copy number within the amplified region.

**b**. Amplification in a 1.2 Mb region on 12p containing five foldback junctions, two inter-chromosomal junctions, and one intra-chromosomal junction. **c**. Amplification in a 6.3 Mb region on 20q containing six foldback junctions. Note the presence of two pairs of adjacent breakpoints flanking the duplications between 55.36 Mb (55.358 Mb and 55.376 Mb) and 55.42 Mb (55.405 Mb and 55.420 Mb) that form long-range translocations. **d**. Amplification of DNA near the end of a broken bridge chromosome by breakage-replication/fusion cycles. In the first cycle, a foldback junction is formed at the chromosome end (filled arrowhead), but a new DSB end is created by replication over an internal ssDNA gap (open arrowhead). If the ssDNA gap is not filled, each round of replication will create a new DSB end (filled arrowhead) and a new SSB end, eventually generating an array of inverted duplications at the original locus of the amplified DNA.

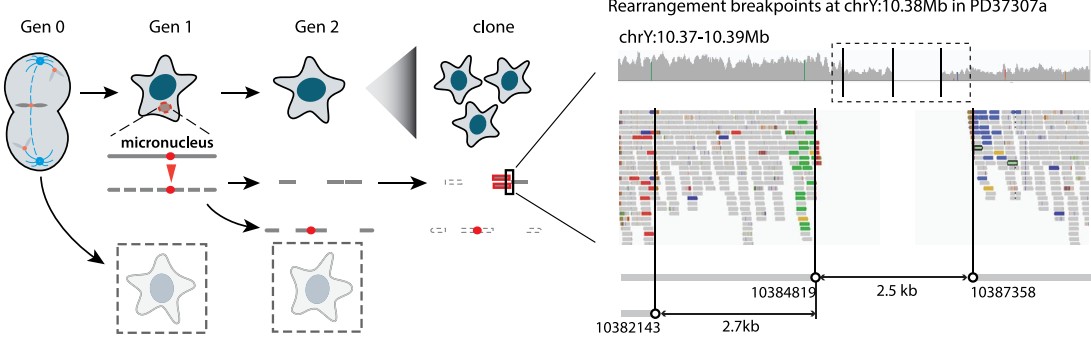

DNA copy-number gains on fragmented chrY from micronuclei

**Extended Data Fig. 3 | See next page for caption.**

**Extended Data Fig. 3 | Chromothripsis and segmental gains of fragmented chrY from micronuclei.** Whole genome sequencing data of clones expanded after induced chrY micronucleation are from Ly et al. (2019). Top: A model of segmental DNA duplications arising by chromosome breakage-replication/fusion. In generation 1 (Gen 1), the chromatid in the micronucleus undergoes poor replication and fragmentation. In generation 2 (Gen 2), a partial reincorporation of the chromosome fragments produces oscillating DNA deletion and retention in both daughters. During subsequent evolution, a single DNA fragment undergoes replication/fusion to create two duplications (red bars) with adjacent parallel breakpoints. An example (from clone PD37307a) expanded after induced Y-chromosome micronucleation is shown on the right. Shown are the total sequence coverage (top) and sequence reads from the region highlighted in the dashed box in the coverage (bottom). The two adjacent parallel breakpoints on the left bound two duplicated segments. A reciprocal breakpoint on the right establishes the origin of all three breakpoints from an ancestral DSB. Bottom: Segmental copy number of chrY in the parental clone and in 12 clones with chrY copy-number alterations. Seven clones that contained intact chrY are not included here (See Supplementary Table 5). chrY copy number is calculated from the total sequence coverage in 10 kb bins (Methods). Gray boxes mask regions containing repetitive sequences where short reads cannot be uniquely mapped. Clones with chromothripsis are annotated with asterisks and clones with segmental copy-number gains are shown in bold. All coordinates are based on alignment to the hs1/T2T CHM13-v2.0 reference. Three examples of adjacent parallel breakpoints are shown on the right. In clone PD37296a, the triplication between 14.4 Mb and 14.6 Mb is bounded by adjacent parallel breakpoints near 14.40 Mb (the two breakpoints on the right are not adjacent); in PD37305a, the duplication between 15.4 and 16.3 Mb is bounded by adjacent parallel breakpoints near 16.26 Mb; in PD37306a, two adjacent breakpoints near 15.75 Mb form a foldback junction. The remnant chrY segments in PD37306a are integrated into a single locus on chr1 at 23.32 Mb, whose structure is shown between the copy-number plot and the breakpoint plot. For the complete list of rearrangement breakpoints and junctions in these clones, see Supplementary Table 6.

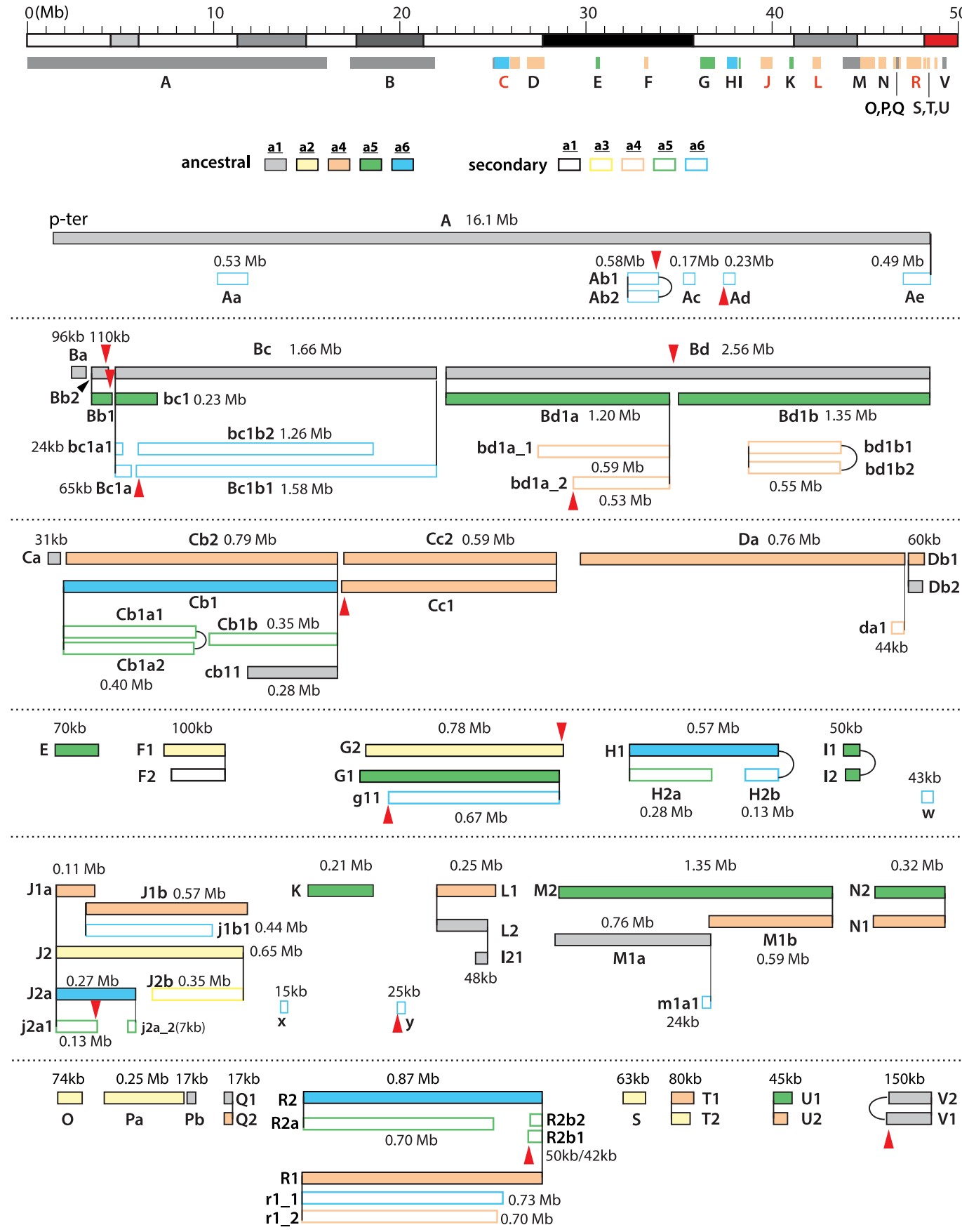

**Extended Data Fig. 4 | See next page for caption.**

**Extended Data Fig. 4 | Rearranged segments of chr4p in bridge clone a.**
The coordinates of each segment (listed in Supplementary Table 10) are
determined from a joint analysis of subclonal copy-number variation
(Supplementary Table 8) and rearrangement junctions (Supplementary Table 9).
Segments are grouped (**A-V**) based on their original locations on 4p (top). Color of
segments: Segments are colored based on the subclone where they are identified
(gray:subclone **a1**; yellow:**a2** and **a3**; orange:**a4**; green:**a5**; blue:**a6**). The same
color scheme is used in Supplementary Table 8 and in Supplementary Note,
Secs. 10 and 11. Ancestral segments are shown as filled bars and secondary
segments are shown as open bars. See below for the definition of ancestral or
secondary segments. Naming of segments: The names of segments reflect both
their original positions (**A** to **V**) and their ancestry. Adjacent segments separated
by small gaps are labelled with 'a','b', etc. (for example, **Ba**-**Bd**). Segments inferred
to have been derived from sister DNA fragments are labelled '1' and '2' with '1'
being the longer segment. Partial duplications (that is, duplicated segments
that are truncated relative to the ancestral segment) have lowercase letters, for
example, **bc** is a truncated copy of **Bc**. When more than one truncated segment is
present, they are distinguished using '_1', '_2', for example, **r1_1**, **r1_2**. Ancestral and
secondary breakpoints: A breakpoint is considered ancestral if (1) it is preserved
in more than one subclone; or (2) it has an adjacent breakpoint that is identified
in a different subclone. A breakpoint is considered secondary otherwise.
Breakpoints shared by more than one segment are highlighted with vertical lines.
Ancestral and secondary segments: A segment is considered ancestral only when
the breakpoints at both boundaries are ancestral. Secondary segments have at
least one boundary being a secondary breakpoint. Identical duplicates of each
segment (that is, with identical breakpoints but located at different locations
on the rearranged chromosome) are shown only once. Arrowheads: Clustered
substitutions reflecting cytosine deamination: downward arrows for deamination
of cytosines on the forward strand; upward arrows for deamination of cytosines
on the reverse strand. Note: (1) Junctions between rearranged segments are not
shown except for foldbacks (arcs) that join adjacent parallel breakpoints [for
example, **Ab1**(-)/**Ab2**(-)]. (2) We infer **H1** to be ancestral and has a sister segment
**H2** that underwent a secondary deletion to produce segment **H2a** (green, retained
in subclone **a5**) and segment **H2b** (blue, joined to **H1** by a foldback junction).
This inference is supported by the structure of the ancestral rearranged
chromosome with **H1/H2** located at the end (see Extended Data Fig. 5).

**a** Ancestral segments that produced rearranged chromosomes in clone a

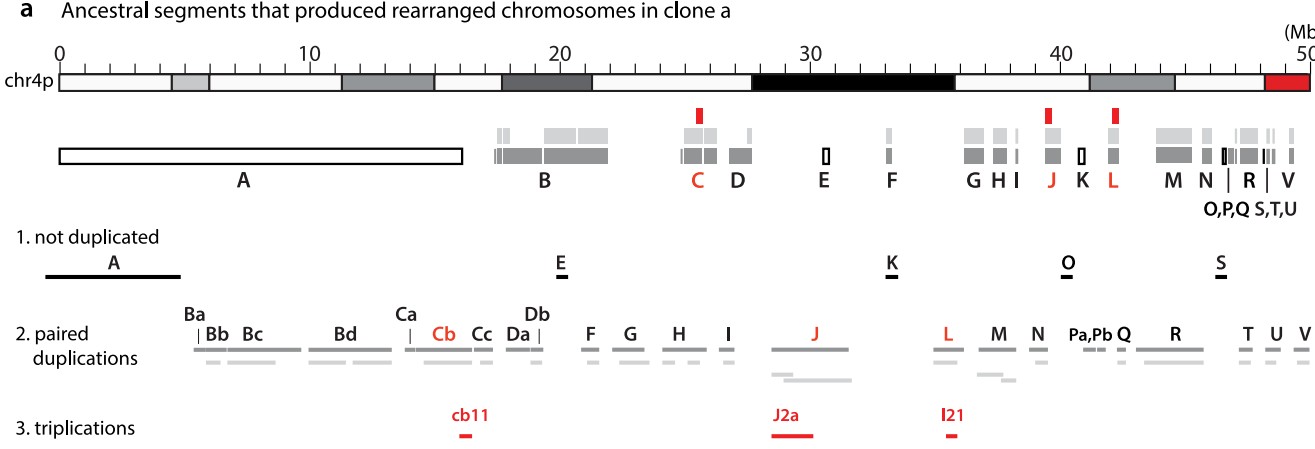

**b** Segmental structure of the rearranged chromosomes in selected subclones

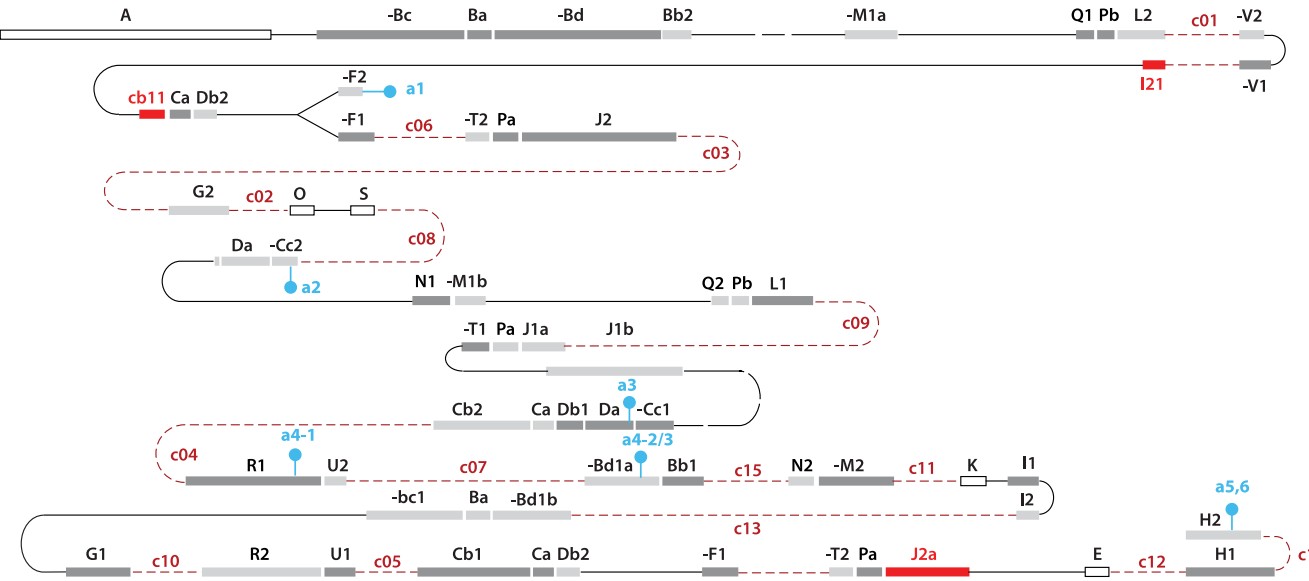

**c** Structure of the ancestral rearranged chromosome highlighting duplications generated by breakage-replication/fusion

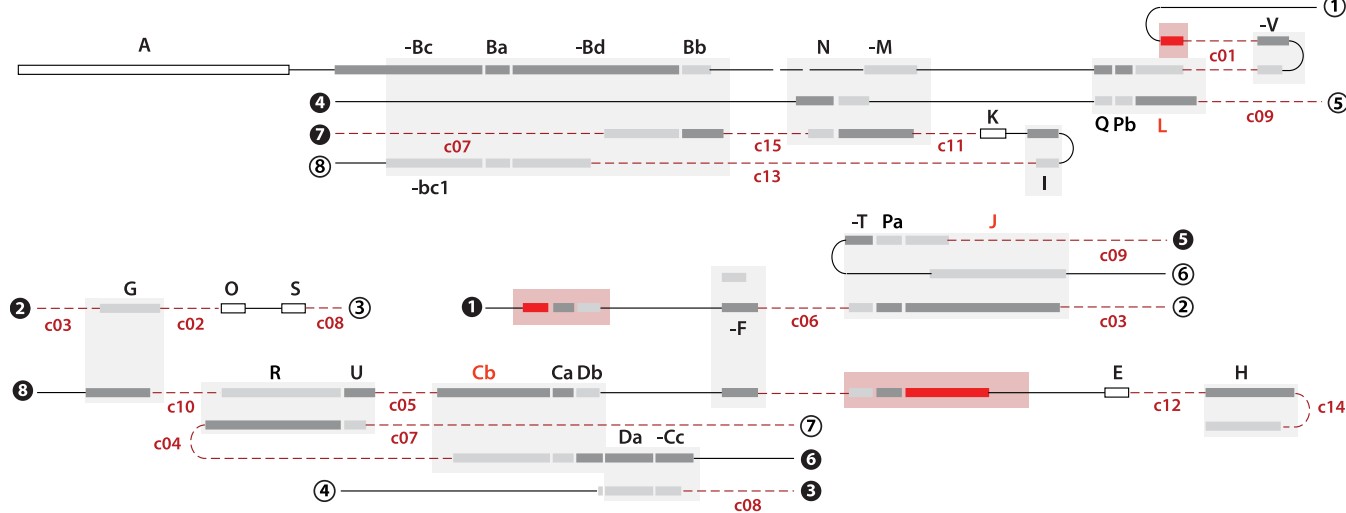

**Extended Data Fig. 5 | See next page for caption.**

**Extended Data Fig. 5 | Segmental structure of rearranged segments and rearranged chromosomes in bridge clone a. a**. Rearranged segments at their original locations. For better visualization, both the size of each segment and the distance between neighbor breakpoints are scaled by a nonlinear function $f(x) = \arctan(x/10000)$. Duplicated segments inferred to be derived from sister DNA fragments generated by breakage-replication/fusion are shown as dark and light gray bars; triplicated segments are shown as red bars. **b**. Joining pattern of rearranged segments in different subclones of clone **a**. The terminus of rearranged chr4 in each subclone is labelled by a blue lollipop. Junctions containing no insertion are shown as long uninterrupted lines; junctions containing one insertion are shown as broken lines; junctions containing two or more insertions are shown as red dashed lines (labelled c1-c15). See Supplementary Table 9 for the complete list of rearrangement junctions and Supplementary Tables 15 and 16 for insertion junctions. **c**. Segmental structure of the ancestral rearranged chromosome highlighting duplications generated by breakage-replication/fusion (segments within shaded areas). Junctions within each shaded area are generated by breakage-fusion-replication; the remaining junctions are generated by breakage-replication-fusion. Numbers denote compound segments that are joined together (for example ① joins ●).

## a   Assessment of breakpoint independence based on breakpoint distance

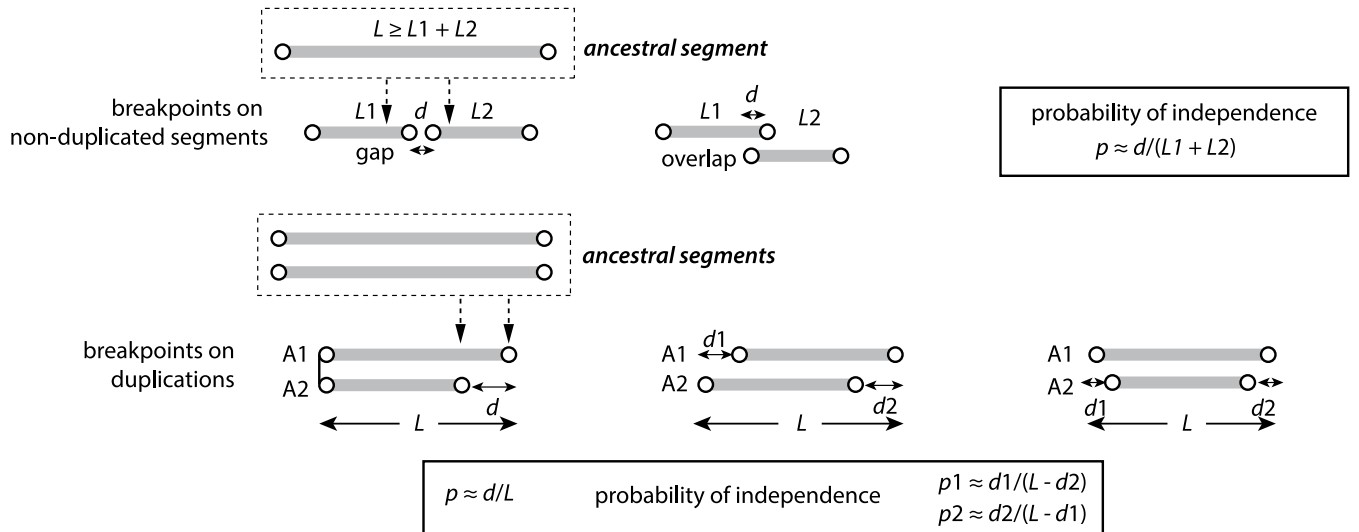

## b   Examples of segments with adjacent gapped and adjacent overlapping breakpoints

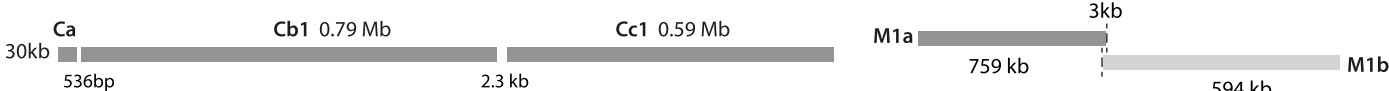

## c   Distance between neighbor breakpoints in bridge clone a

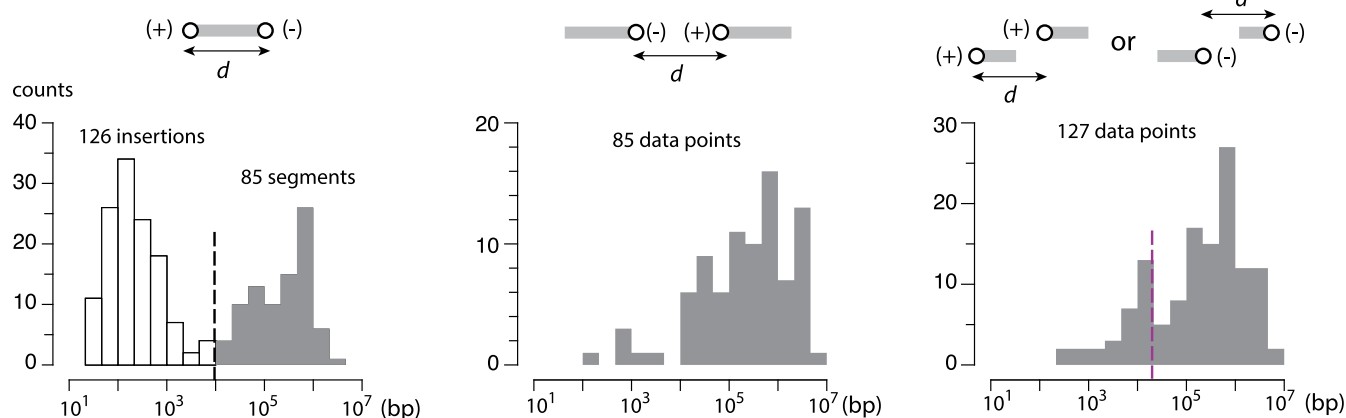

**Extended Data Fig. 6 | Identification of adjacent breakpoints in bridge clone a based on breakpoint distance. a.** Assessment of the likelihood that two breakpoints are generated independently based on the ratio of breakpoint distance relative to the size of the ancestral segment where breakpoints are derived. For gapped or overlapping breakpoints, the size of the ancestral segment ($L$) has to be equal to or larger than the combined size of the two adjacent segments ($L1 + L2$). The probability that two breakpoints within distance $d$ are generated independently is approximately $d/L \leq d/(L1 + L2)$. Examples are shown in **b**. For parallel breakpoints located on two duplicated segments, the size of the ancestral segment has to be equal to or larger than the longer segment. The probability that two parallel breakpoints within distance $d$ are generated

independently is approximately $d/L$, where $L$ is the size of the longer segment. **b.** Examples of three adjacent segments separated by small gaps (left) and two adjacent segments with a small overlap (right) identified in clone **a**. **c.** Distance distributions of breakpoints in three configurations in clone **a**. Left: small insertions (open bars) and large segments (filled). The only segment longer than 10 Mb is omitted from the histogram. Most short insertions are within 1 kb and the threshold is set to be 10 kb. Middle: distance between nearest gapped breakpoints. Most adjacent gapped breakpoints are within 10 kb. Right: distance between nearest parallel breakpoints. The threshold for adjacent parallel breakpoints (20 kb) is shown as a red dashed line.

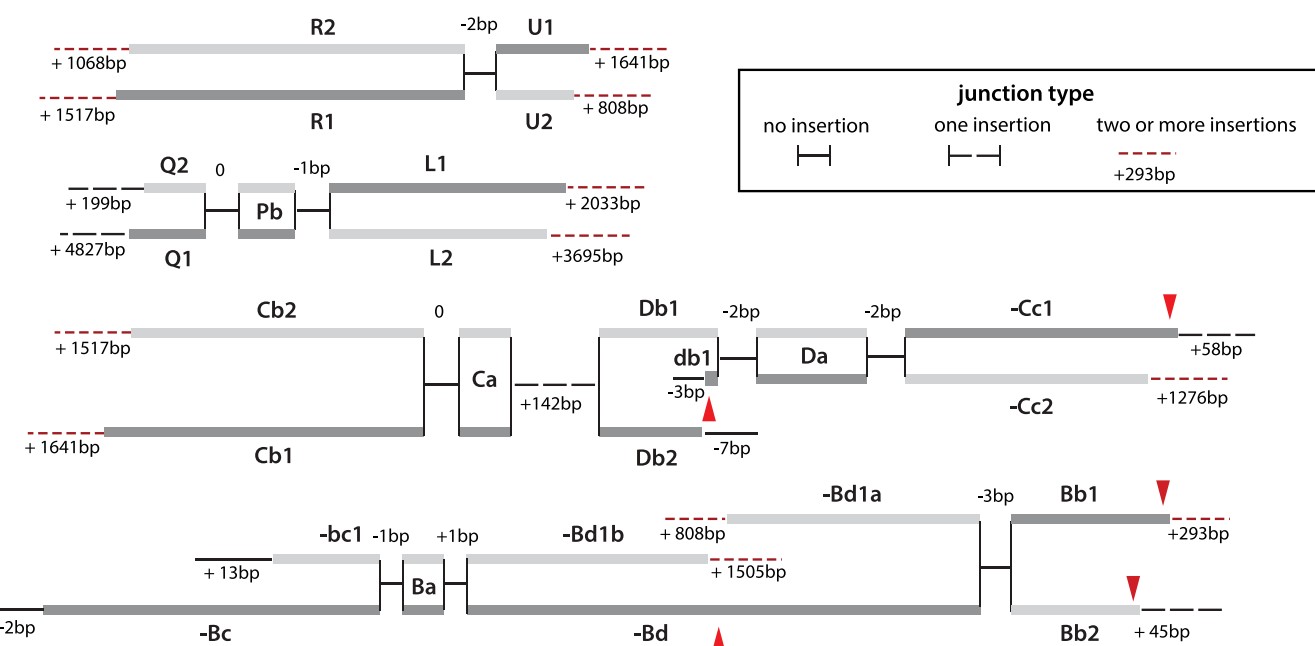

**a Sister duplications generated by B-R/F**

**b**

**c Strand coordination between breakpoints and clustered substitutions**

**Extended Data Fig. 7 | See next page for caption.**

**Extended Data Fig. 7 | Additional genomic evidence of breakage-replication/ fusion in bridge clone a. a.** Inferred evolutionary history of duplicated segments from region **B**:17.4-21.7 Mb, **C**:24.9-26.6 Mb, and **D**:26.8-27.6 Mb of chr4p. In the top three rows, dark gray bars indicate ancestral DNA fragments (**Ba-Bd**, **Ca-Cc**, **Da,Db**) and the outcome of fusions before replication ('-' for inverted segments). In the fourth row, dark and light gray bars represent sister DNA fragments derived from complementary strands of the ancestral DNA by replication. For the terminal segments (**Bc**, **Bb**, **Cc**, **Cb**), dark gray bars indicate segments containing the ancestral 3' overhang and light gray bars indicate segments containing the shorter 5'-ends. The pairing of ancestral 3' and 5'-ends is highlighted in dashed boxes. For the internal segments, dark or light gray are arbitrary. The bottom row shows the final structure of the rearranged DNA and their copy-number states in each subclone. Although each clone only contains a subset of these segments, the proximity between breakpoints indicates that they are all ancestral. Note the prevalence of complex insertions (red dashed lines) at breakage-replication-fusion junctions between sister DNA fragments. **b.** Clustered substitutions at the right end of the **Bb2** segment indicating deamination of newly synthesized DNA. Mutations indicating cytosine deamination on the forward DNA strand (C > T substitutions). As **Bb2** originates from reverse strand DNA, deamination of forward strand DNA must have occurred after replication and secondary resection. **c.** Strand coordination of mutations on compound duplications generated by breakage-replication/fusion. Sister segments are shown as dark and light gray bars following the convention in **a**. Shading for segments bounded by flush breakpoints (vertical lines) is arbitrary. Linkage between segments (dark and light gray bars), for example, **R2**–**U1** and **R1**–**U2**, is determined from the rearranged chromosome as shown in Extended Data Fig. 5. Strand coordination is observed for (1) signature of deamination; and (2) breakpoints, that is, the shorter breakpoint on one side of a segment (ancestral 5'-end) is paired with the longer breakpoint on the opposite side (ancestral 3'-end). Both types of strand coordination support the conclusion that sister segments are derived from complementary DNA strands of a single ancestral dsDNA fragment. Note that the red arrows are oriented relative to the segment where deamination occurred. Therefore, if the segment is inverted (for example, **-Cc1** and **-Bd**), the signature of deamination is also flipped. The deamination signature on the **db1** segment is discussed in detail in Supplementary Note, Sec. 3. Two additional examples of compound sister duplications are discussed in Extended Data Fig. 8. Sequence features of junctions between segments: '-2bp' for 2 bp microhomology; '+142 bp' for an insertion of 142 bp; for junctions containing two or more insertions, only the total length of the inserted sequence is shown.

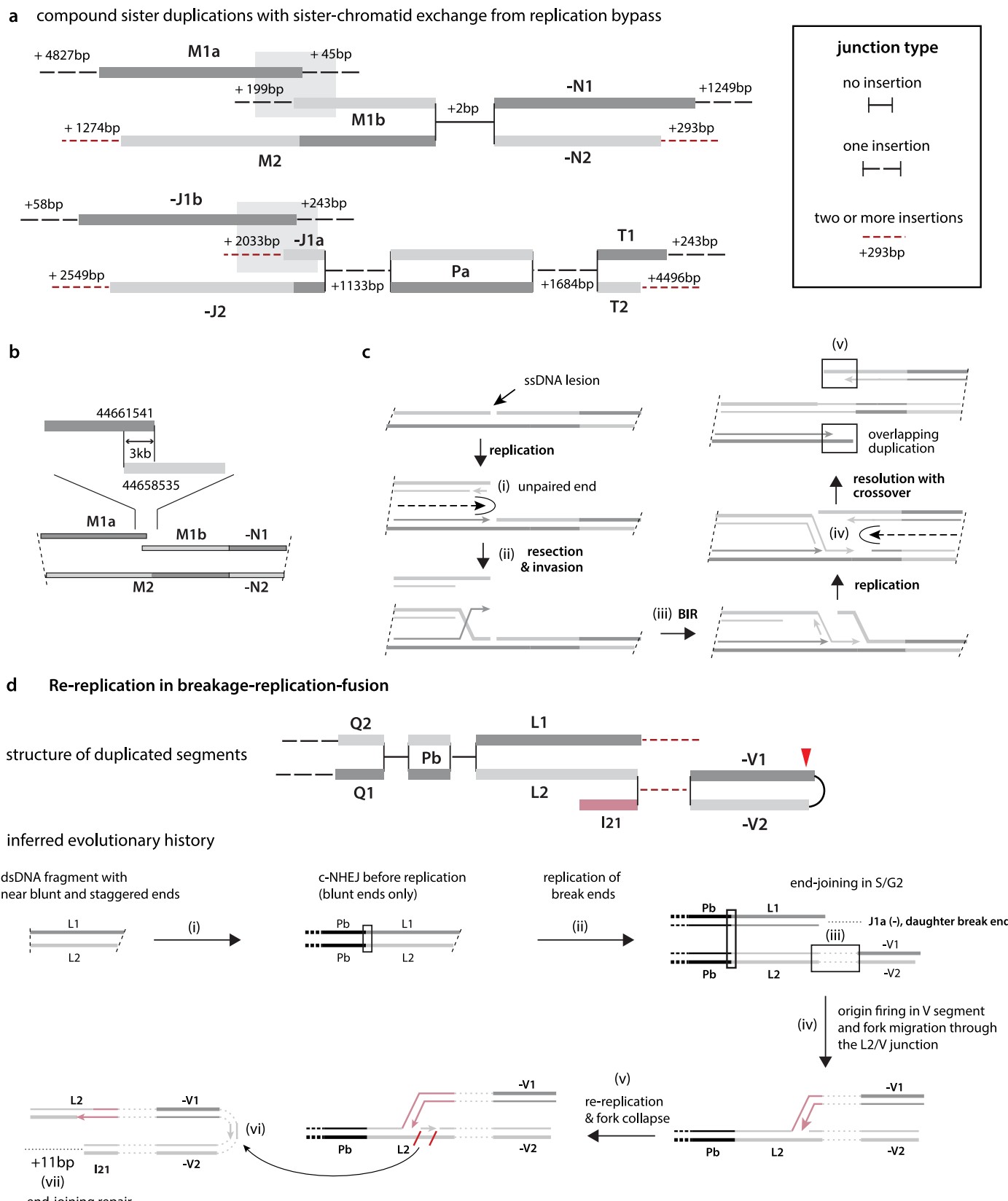

**Extended Data Fig. 8 | See next page for caption.**

**Extended Data Fig. 8 | Two processes of breakage-replication/fusion that result in DNA over-replication. a**. Two instances of segments with a small overlap (shaded areas) accompanied with opposite coordination between distal breakpoints. The **M1a/M1b** segments are unambiguously determined from subclonal copy-number variation. The configuration of **J1a/J1b** segments is inferred from the strand coordination of breakpoints. We suggest that both instances are generated by the replication bypass mechanism that causes sister-chromatid exchange. **b**. Structure of the **M1a,M1b/M2** segments and their neighbor segments **N1/N2** in the rearranged chromosomes. **c.** A proposed model for the generation of **M1a**, **M1b** and **M2** by replication bypass. Thick lines represent template DNA strands; thin lines represent newly synthesized DNA; arrowheads denote the 3′-ends of newly synthesized DNA. DNA strands are colored similarly as in **b**. (i) A right-moving fork (dashed arrow) collapses (for example, at a ssDNA gap) and creates an unpaired dsDNA end (light gray). (ii) The dsDNA end (light gray) is resected and invades the sister DNA template (dark gray). (iii) The invading end initiates displacement DNA synthesis that has a similar outcome as break-induced replication (BIR) but may be carried out by different enzymes. (iv) A left-moving fork encounters the BIR fork (light gray) and collapses. (v) A resolution involving cleavage of the bottom dark gray strand causes sister-DNA exchange (dark gray strand joining light gray strand),

explaining the opposite coordination between breakpoints shown in **a**. Finally, the two ends are ligated to other DNA ends to form two translocation junctions. See Supplementary Note, Sec. 13 for examples of overlapping translocations from a different cancer genome (K-562) and Supplementary Table 21 for examples from the HCC1954 genome. **d**. An example of triplication indicating DNA re-replication caused by a fusion between replicated and unreplicated DNA. Top: Structure of duplications of the **L** segments. Bottom: Inferred evolutionary history that involves DNA re-replication. We infer that the small duplication **l21** was generated by a replisome initiated from the **V** fragment migrating into an already replicated **L2** segment (i-iii). Following the firing of replication origins within the **V** segment, bidirectional fork progression first generated the sister segments **V1/V2**, then duplicated the **L2:-V** junction, and further caused partial re-replication of **L2** (iv and v). As the **L2** was already replicated, the replication fork passing into the **L2** segment could not merge with an opposite fork: This replication fork eventually collapsed (vi), creating an unpaired DSB end on **l21** that was then ligated to another unpaired DNA end on the **cb11** segment. The presence of an insertion within the foldback junction between **V1/V2** that is mapped to chr4:42,275,638-873 right next to the breakpoint of **l21** at 42,275,875 further supports that steps (v) and (vi) occur simultaneously. A similar model may explain the triplicated **cb11** segment (Supplementary Note, Sec. 10).

**a** Clustered origins of short insertions in bridge clone 1a

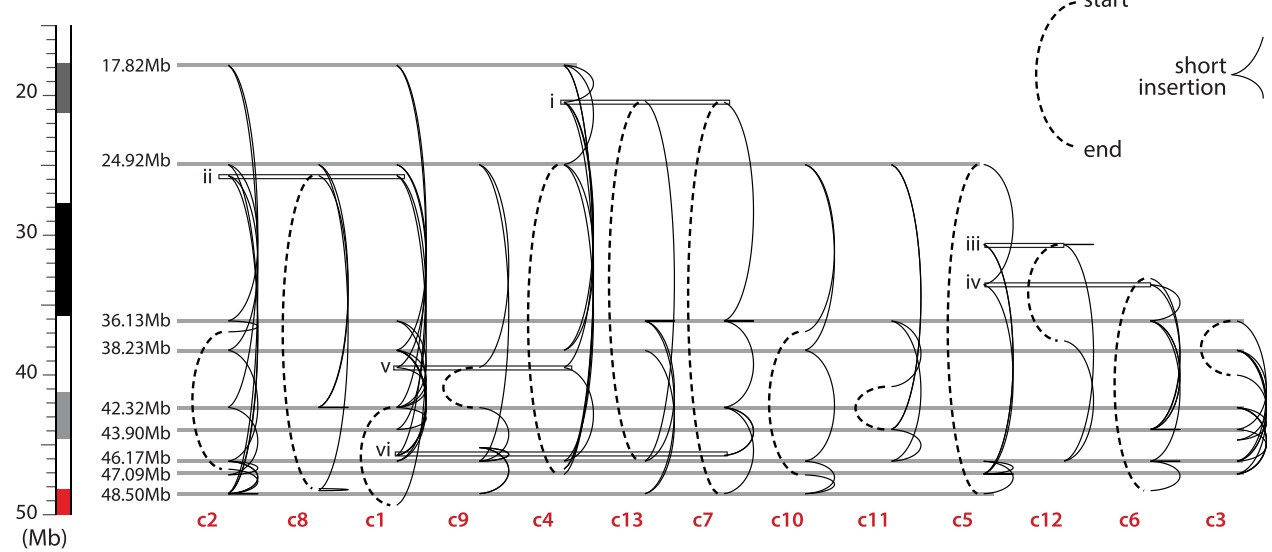

**b** Clustered origins of insertions in the X-29 clone
from Maciejowski et al. (2015)

**c** Tiling pattern of insertions in a primary cancer
from Achom, Sadagopan, Bao et al. (2024)

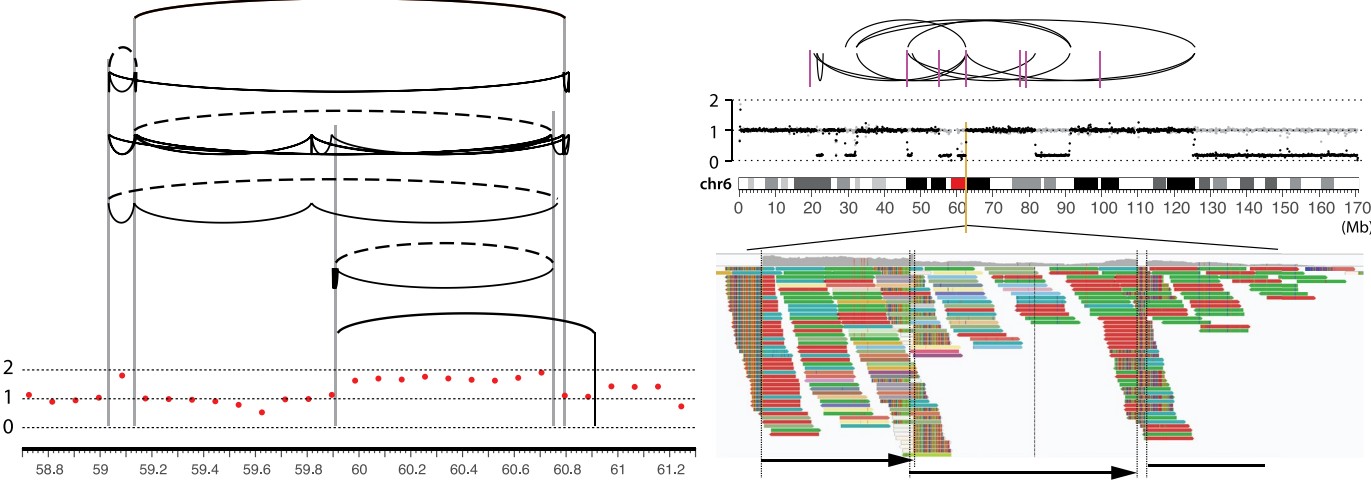

**d** Tiling pattern of insertions near chr1:151Mb in the HCC1954 genome

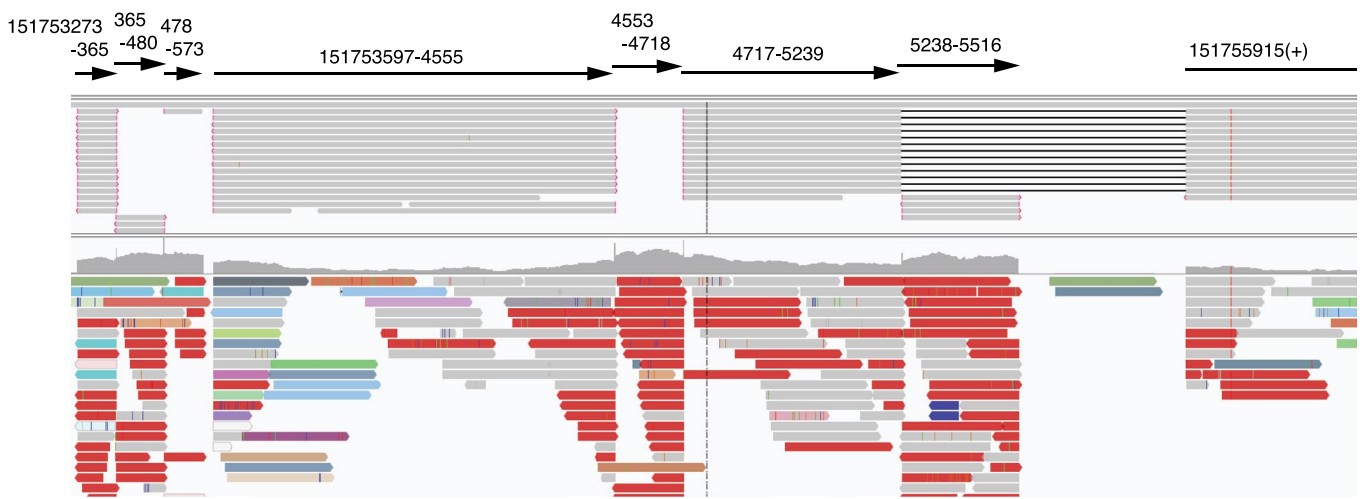

**Extended Data Fig. 9 | See next page for caption.**

**Extended Data Fig. 9 | Additional data of short insertions with a tiling pattern at the origin sites. a**. Footprints of the 13 chains of insertions (c1-c13) as shown in Fig. 5c. For each chain, the beginning and the end are linked by a dashed line, solid arcs represent insertions that are strung together between these breakpoints. In addition to 9 hotspots shown in Fig. 5b, there are six regions each contributing two or three insertions: i. 20.48 Mb; ii. 25.72 Mb; iii. 30.66 Mb; iv. 33.54 Mb; v. 39.47 Mb; vi. 45.76 Mb. **b**. Four complex insertion junctions identified in a post-crisis RPE-1 clone. Red dots show haplotype-specific coverage in 90 kb bins. Chained insertions are shown similarly as in **a** except that the chromosome is drawn horizontally. **c**. Two insertions (arrowheads) adjacent to a rearrangement breakpoint identified on chr6 in a translocation renal cell carcinoma from Ref. 30. Black and gray dots show haplotype-specific coverage in 25 kb bins. **d**. Seven insertions mapped to a region on chr1q near the breakpoint chr1:151,755,915(+) in the HCC1954 genome resolved by long reads (top) and short reads (bottom). For additional data on insertions in the HCC1954 genome, see Supplementary Note, Sec. 7 and Supplementary Table 20.

**a** Insertion and overlapping breakpoints

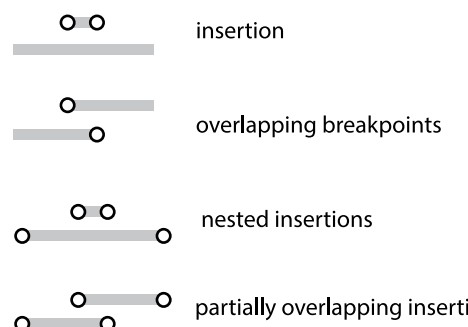

**c** Four-breakpoint footprints generated by one breakage-replication/fusion cycle

**b** Breakpoints generated by two or more cycles of breakage-replication/fusion

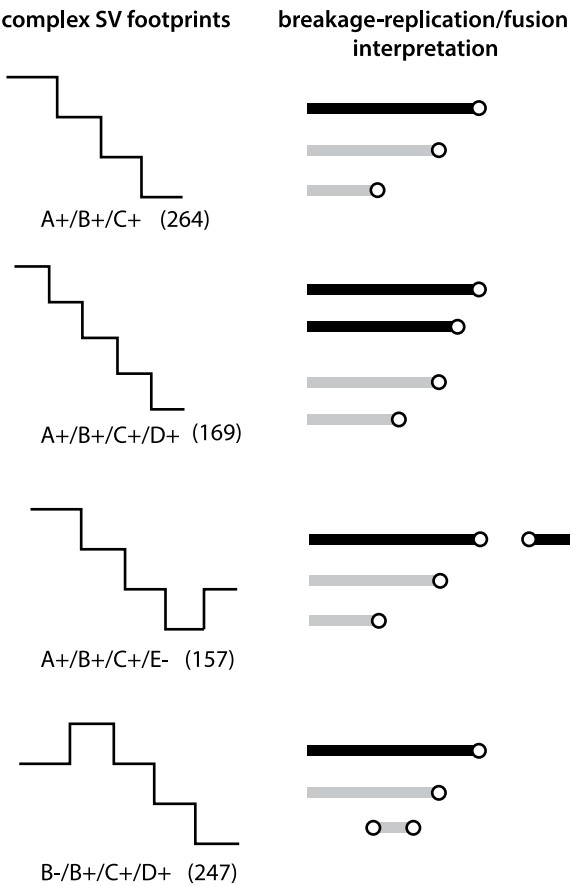

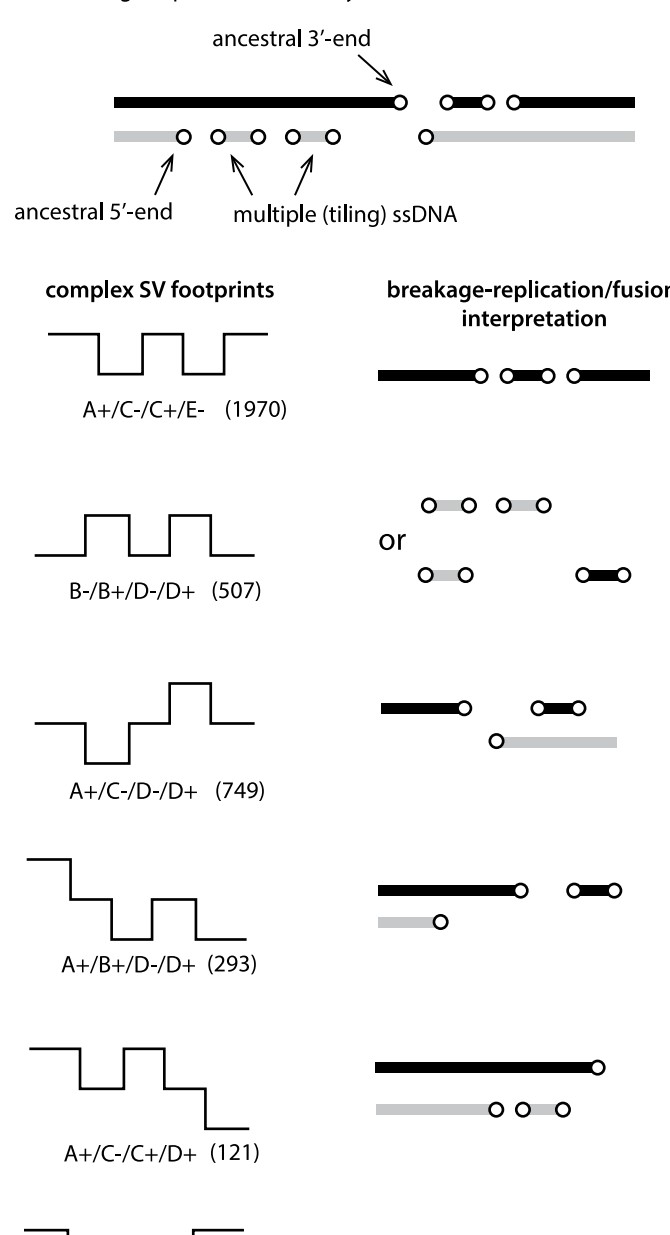

**Extended Data Fig. 10 | Patterns of complex rearrangement footprints in cancer genomes that may be explained by breakage-replication/fusion.**
**a**. Ambiguity between a short insertion and two adjacent overlapping breakpoints (top), or between two partially overlapping insertions and two nested insertions (bottom). If the distance between the two inner breakpoints exceeds the maximum size of sequencing reads, then the two configurations cannot be distinguished. In our analysis of adjacent breakpoints in the cancer genomes with only shotgun sequencing data, we treat overlapping breakpoints as 'insertions' and treat both nested insertions and partially overlapping insertions as overlapping insertions. **b**. Footprints of three or four rearrangement

breakpoints identified in Ref. 16 that can be explained by rearrangements generated over two breakage-replication/fusion cycles. Where there are two gray segments or two black segments, they are derived from a second cycle. **c**. Four-breakpoint footprints that can arise from one breakage-replication/fusion cycle with one or multiple insertions derived from one or both DSB ends. In **b** and **c**, numbers in parentheses represent the total counts of instances of each footprint, regardless of the joining pattern between breakpoints. For the original report of the complex rearrangement footprints and the number of instances, see pages 63-83 of the Supplementary Information of Ref. 16.

# Reporting Summary

## Statistics

For all statistical analyses, confirm that the following items are present in the figure legend, table legend, main text, or Methods section.

| n/a | Confirmed | |
|---|---|---|
| ☒ | ☐ | The exact sample size ($n$) for each experimental group/condition, given as a discrete number and unit of measurement |
| ☒ | ☐ | A statement on whether measurements were taken from distinct samples or whether the same sample was measured repeatedly |
| ☒ | ☐ | The statistical test(s) used AND whether they are one- or two-sided<br>*Only common tests should be described solely by name; describe more complex techniques in the Methods section.* |
| ☒ | ☐ | A description of all covariates tested |
| ☒ | ☐ | A description of any assumptions or corrections, such as tests of normality and adjustment for multiple comparisons |
| ☒ | ☐ | A full description of the statistical parameters including central tendency (e.g. means) or other basic estimates (e.g. regression coefficient) AND variation (e.g. standard deviation) or associated estimates of uncertainty (e.g. confidence intervals) |
| ☒ | ☐ | For null hypothesis testing, the test statistic (e.g. $F$, $t$, $r$) with confidence intervals, effect sizes, degrees of freedom and $P$ value noted<br>*Give P values as exact values whenever suitable.* |
| ☒ | ☐ | For Bayesian analysis, information on the choice of priors and Markov chain Monte Carlo settings |
| ☒ | ☐ | For hierarchical and complex designs, identification of the appropriate level for tests and full reporting of outcomes |
| ☒ | ☐ | Estimates of effect sizes (e.g. Cohen's $d$, Pearson's $r$), indicating how they were calculated |

*Our web collection on statistics for biologists contains articles on many of the points above.*

## Software and code

Policy information about availability of computer code

| Data collection | No special software was used for data collection. All data used in the current study are publicly available from the Short Read Archive. |
|---|---|
| Data analysis | Script for the identification of adjacent breakpoints is uploaded to the github repository (https://github.com/chengzhongzhangDFCI/BRF). No special software or code were used for data processing or analysis. Data processing and analytical workflows were described in Methods with references to prior publications. |

For manuscripts utilizing custom algorithms or software that are central to the research but not yet described in published literature, software must be made available to editors and reviewers. We strongly encourage code deposition in a community repository (e.g. GitHub). See the Nature Portfolio guidelines for submitting code & software for further information.

## Data

Policy information about availability of data

All manuscripts must include a data availability statement. This statement should provide the following information, where applicable:
- Accession codes, unique identifiers, or web links for publicly available datasets
- A description of any restrictions on data availability
- For clinical datasets or third party data, please ensure that the statement adheres to our policy

Data availability is included in the data availability statement at the end of the main text.

## Research involving human participants, their data, or biological material

Policy information about studies with human participants or human data. See also policy information about sex, gender (identity/presentation), and sexual orientation and race, ethnicity and racism.

| | |
|---|---|
| Reporting on sex and gender | This study does not involve human participants. So this is not applicable. |
| Reporting on race, ethnicity, or other socially relevant groupings | Not applicable. |
| Population characteristics | Not applicable. |
| Recruitment | Not applicable. |
| Ethics oversight | Not applicable |

Note that full information on the approval of the study protocol must also be provided in the manuscript.

# Field-specific reporting

Please select the one below that is the best fit for your research. If you are not sure, read the appropriate sections before making your selection.

☒ Life sciences        ☐ Behavioural & social sciences        ☐ Ecological, evolutionary & environmental sciences

For a reference copy of the document with all sections, see nature.com/documents/nr-reporting-summary-flat.pdf

# Life sciences study design

All studies must disclose on these points even when the disclosure is negative.

| | |
|---|---|
| Sample size | All sample sizes are explicitly described as needed. |
| Data exclusions | No data were excluded. |
| Replication | Replication of findings using other prior published data is explicitly stated. |
| Randomization | Not applicable. |
| Blinding | Not applicable. |

# Reporting for specific materials, systems and methods

We require information from authors about some types of materials, experimental systems and methods used in many studies. Here, indicate whether each material, system or method listed is relevant to your study. If you are not sure if a list item applies to your research, read the appropriate section before selecting a response.

### Materials & experimental systems

| n/a | Involved in the study |
|---|---|
| ☒ ☐ | Antibodies |
| ☒ ☐ | Eukaryotic cell lines |
| ☒ ☐ | Palaeontology and archaeology |
| ☒ ☐ | Animals and other organisms |
| ☒ ☐ | Clinical data |
| ☒ ☐ | Dual use research of concern |
| ☒ ☐ | Plants |

### Methods

| n/a | Involved in the study |
|---|---|
| ☒ ☐ | ChIP-seq |
| ☒ ☐ | Flow cytometry |
| ☒ ☐ | MRI-based neuroimaging |

## Plants

| | |
|---|---|
| Seed stocks | Not applicable |
| Novel plant genotypes | Not applicable |
| Authentication | Not applicable |

