## [Peer Review File · Nature Genetics]

A breakage-replication/fusion process explains complex rearrangements and segmental DNA amplification

Corresponding Author: Dr Cheng-Zhong Zhang

This manuscript has been previously reviewed at another journal. This document only contains information relating to versions considered at Nature Genetics.

Version 0:

Decision Letter:

Our ref: NG-A69553-T

11th Aug 2025

Dear Dr Zhang,

Thank you for submitting your revised manuscript "A breakage-replication/fusion process generating complex rearrangements and segmental DNA amplification" (NG-A69553-T). It has now been seen by the original referees and their comments are below. The reviewers find that the paper has improved in revision, and therefore we'll be happy in principle to publish it in Nature Genetics, pending minor revisions to comply with our editorial and formatting guidelines.

Sincerely,

Safia Danovi, PhD
Senior Editor, Nature Genetics
ORCID: 0009-0007-7822-5479

Reviewer #1 (Remarks to the Author):

The authors have significantly rewritten the manuscript and figures for further clarity. Though it is still a technically challenging topic due to the inherent complexity of the biological process being studied, there are sufficient resources that the authors included for the interested reader to understand the paper. I have no further comments, and I believe the paper is eminently suitable for the audience of Nature Genetics.

Reviewer #2 (Remarks to the Author):

The authors have addressed the concerns raised in my initial review. Notably, their clarification of how the B-R/F model accounts for adjacent parallel breakpoints via double-strand break (DSB) end replication represents a meaningful conceptual advance over the lesion segregation model. This mechanistic distinction further emphasizes the conceptual novelty of the current manuscript and provides important insight into interpreting the mechanisms of structural variation in human cancer genomes. I am therefore in support of publication.
